# PRC1 sustains the integrity of neural fate in the absence of PRC2 function

**Ayana Sawai[1], Sarah Pfennig[1], Milica Bulajić[2], Alexander Miller[1], Alireza Khodadadi-Jamayran[3], Esteban O Mazzoni[2], Jeremy S Dasen[1]***

[1]Neuroscience Institute, Department of Neuroscience and Physiology, NYU School of Medicine, New York, United States; [2]Department of Biology, New York University, New York, United States; [3]Applied Bioinformatics Laboratories, Office of Science and Research, NYU School of Medcine, New York, United States

**Abstract** Polycomb repressive complexes (PRCs) 1 and 2 maintain stable cellular memories of early fate decisions by establishing heritable patterns of gene repression. PRCs repress transcription through histone modifications and chromatin compaction, but their roles in neuronal subtype diversification are poorly defined. We found that PRC1 is essential for the specification of segmentally restricted spinal motor neuron (MN) subtypes, while PRC2 activity is dispensable to maintain MN positional identities during terminal differentiation. Mutation of the core PRC1 component *Ring1* in mice leads to increased chromatin accessibility and ectopic expression of a broad variety of fates determinants, including *Hox* transcription factors, while neuronal class-specific features are maintained. Loss of MN subtype identities in *Ring1* mutants is due to the suppression of Hox-dependent specification programs by derepressed *Hox13* paralogs (*Hoxa13*, *Hoxb13*, *Hoxc13*, *Hoxd13*). These results indicate that PRC1 can function in the absence of de novo PRC2-dependent histone methylation to maintain chromatin topology and postmitotic neuronal fate.

## Editor's evaluation

This is an exciting and very well executed study, which will be of broad interest to the field of neuronal development. The demonstration of a similar logic in mouse as to what was reported earlier for *Drosophila* PRCs supports the idea of a deeply conserved mechanism of rostrocaudal patterning where PRC complexes control Hox gene expression.

***For correspondence:** jeremy.dasen@nyumc.org

**Competing interest:** The authors declare that no competing interests exist.

## Introduction

Accurate control of gene expression is essential for the specification and maintenance of neural fates during development. Studies of cell-type-restricted transcription factors have illuminated the mechanisms by which spatial and temporal regulation of gene expression gives rise to identifiable neuronal subtypes (*Doe, 2017*; *Fishell and Kepecs, 2020*; *Hobert and Kratsios, 2019*; *Sagner and Briscoe, 2019*; *Venkatasubramanian and Mann, 2019*). A parallel and critical mechanism of gene regulation is through the post-translational modification of histones, which enables and restricts transcription by modifying chromatin structure (*Kishi and Gotoh, 2018*; *Schuettengruber et al., 2017*). The Polycomb group (PcG) is key family of histone-associated proteins that play evolutionarily conserved roles in restricting gene expression during development (*Blackledge et al., 2015*; *Gentile and Kmita, 2020*; *Simon and Kingston, 2009*; *Soshnikova and Duboule, 2009*). In embryonic stem cells, cell fate determinants are repressed through PRC activities, and PRC-associated histone marks are subsequently removed from loci as cells differentiate (*Boyer et al., 2006*; *Farcas et al., 2012*; *Tavares et al., 2012*). PRC repression is maintained through cell division and after differentiation and is thought to

contribute to stable cellular memories of early patterning events (*Ciabrelli et al., 2017*; *Coleman and Struhl, 2017*). In vertebrates, much of our knowledge of how PRCs regulate gene expression has emerged from biochemical studies of PcG proteins or from the activity of these factors in proliferating cells. Despite an in depth understanding of the mechanisms of PRC action, how PcG proteins interact with gene regulatory networks in the CNS remains poorly understood.

The specification of neuronal fates in the vertebrate spinal cord provides a tractable system to elucidate the function of PcG proteins, as the pathways that determine identities are well-defined, and the molecular signatures of many subtypes are known (*Butler and Bronner, 2015*; *Sagner and Briscoe, 2019*). One neuronal class where fate specification has been closely examined is the spinal MN. A core set of transcription factors, including Mnx1, Isl1/2, and Lhx3, determines class-specific features of MNs, including neurotransmitter identity (*Shirasaki and Pfaff, 2002*). The subsequent diversification of MNs into hundreds of muscle-specific subtypes is achieved through a conserved network of Hox transcription factors differentially expressed along the rostrocaudal axis (*Philippidou and Dasen, 2013*). During neural tube patterning, opposing gradients of retinoic acid (RA) and fibroblast growth factors (FGFs) provide spinal progenitors with a positional identity (*Bel-Vialar et al., 2002*; *Dasen et al., 2003*; *Liu et al., 2001*). These morphogens act, in part, by temporally and spatially depleting Polycomb-associated histone marks from *Hox* clusters (*Mazzoni et al., 2013*). As progenitors exit the cell cycle, MNs continue to express *Hox* genes, where they regulate repertoires of subtype-specific genes (*Catela et al., 2016*; *Dasen et al., 2005*; *Mendelsohn et al., 2017*). Although the role of Hox proteins in the CNS are well-characterized (*Parker and Krumlauf, 2020*), and are known targets of PRC activities (*Gentile and Kmita, 2020*), the specific contributions of PRC1 and PRC2 to CNS maturation remain unclear, as few studies have directly compared their functions during embryonic development.

Polycomb repression is initiated by PRC2, which methylates histone H3 at lysine-27 (H3K27me3), permitting recruitment of PRC1 through subunits that recognize this mark, leading to chromatin compaction at genes targeted for repression (*Margueron and Reinberg, 2011*; *Schuettengruber et al., 2017*). In embryonic stem (ES) cells, *Hox* gene clusters are initially covered by H3K27me3 and loss of PRC2 function leads to reduced PRC1 binding and ectopic *Hox* expression (*Boyer et al., 2006*). During mouse development, H3K27me3 marks are progressively removed from *Hox* clusters, allowing for the temporal and spatial activation of more caudal *Hox* genes during axis extension (*Soshnikova and Duboule, 2009*). The progressive removal of PRC2-associated histone marks is also recapitulated in ES cell-derived MNs (ESC-MNs), where RA functions to deplete H3K27me3 from rostral *Hox1-Hox5* paralogs, while FGF removes H3K27me3 from more caudal *Hox* genes through Cdx proteins (*Mazzoni et al., 2013*). Although loss of PRC2 affects the viability of ESC-MNs, a hypomorphic mutation in the PRC2 component *Suz12* leads to ectopic *Hox* expression (*Mazzoni et al., 2013*). Thus, during early phases of embryonic development, PRC2 appears to have a critical role in establishing the early profiles of *Hox* expression in MNs along the rostrocaudal axis.

PRC1 and PRC2 can also exist in a variety of configurations, which may contribute to neuronal subtype-specific activities. The core subunit of PRC1, Ring1, binds to one of six Polycomb group Ring finger (Pcgf) proteins (*Gao et al., 2012*). PRC1 containing Pcgf4 interacts with Cbx proteins (canonical PRC1) which recognize H3K27me3 (*Bernstein et al., 2006*; *Morey et al., 2012*). Variant forms of PRC1 containing Rybp can inhibit incorporation of Cbx proteins into PRC1, and bind target loci independent of H3K27me3 (*Tavares et al., 2012*; *Wang et al., 2010*). We previously found that a PRC1 component, Pcgf4 (Bmi1), is required to establish rostral boundaries of *Hox* expression in differentiating MNs (*Golden and Dasen, 2012*). By contrast, deletion of a core PRC2 component, *Ezh2*, from MN progenitors had no apparent impact on fate specification, possibly due to compensation by *Ezh1* (*Shen et al., 2008*), or through use of variant PRC1 containing Rybp. These observations raise the question of what are the specific roles of canonical PRC1, variant PRC1, and PRC2 during MN subtype diversification.

To determine the function of PRCs during neural differentiation, we removed core components of these complexes from MN progenitors. We found that depletion of Ring1 proteins, essential constituents of all PRC1 complexes, causes pronounced changes in transcription factor expression and a loss of Hox-dependent MN subtypes. By contrast, neither PRC2 nor variant PRC1 activities are required to maintain rostrocaudal positional identities at the time of MN differentiation. Deletion of *Ring1* leads to increased chromatin accessibility and derepression of a broad variety of cell fate determinants, while

class-specific features of MNs are preserved. The derepression of caudal *Hox* genes in *Ring1* mutants leads to the suppression of MN subtype diversification programs. These findings indicate that PRC1 function is essential during terminal differentiation to specify the transcriptional identities of motor neurons.

## Results

### PRC1 is essential for rostrocaudal patterning during neuronal differentiation

To determine the relative contributions of PRC1 and PRC2 to neuronal specification, we analyzed mice in which core subunit-encoding genes were selectively removed from MN progenitors. *Ezh* genes encode the methyltransferase activity of PRC2, while *Eed* is required to enhance this function. We first generated mice in which both *Ezh* genes are conditionally deleted, by breeding *Ezh1* and *Ezh2* floxed lines to *Olig2^Cre* mice, which targets Cre to MN progenitors (Ezh^MNΔ mice) (**Hidalgo et al., 2012**; **Su et al., 2003**). We confirmed MN-restricted loss of PRC2 activity by examining the pattern of H3K27me3, which was selectively depleted from progenitors and post-mitotic MNs by E11.5 (**Figure 1A and B**, **Figure 1—figure supplement 1A,B**). Expression of Mnx1 and *Slc18a3* (*Vacht*), two general markers of MN identity, were grossly unchanged in Ezh^MNΔ mice (**Figure 1C, D and G**). We next analyzed expression of Hox proteins in Ezh^MNΔ mice and found that the MN columnar subtype determinants Hoxc6, Hoxc9, and Hoxc10, were all expressed in their normal domains (**Figure 1E and F**, **Figure 1—figure supplement 1C-E**).

Removal of *Ezh* genes via *Olig2^Cre* depletes PRC2 function in MN progenitors, raising the possibility that PRC2 regulates *Hox* expression at earlier stages. We therefore analyzed mice in which the core PRC2 component *Eed* was deleted using *Sox1^Cre* (Eed^NEΔ mice), which targets Cre to neuroectoderm (**Takashima et al., 2007**; **Yaghmaeian Salmani et al., 2018**). In Eed^NEΔ mice, H3K27me3 was depleted from spinal progenitors and postmitotic neurons by E11.5, with some H3K27me3 present in the floor plate (**Figure 1—figure supplement 1G**). In Eed^NEΔ mice, the number of Mnx1+ cells, and pattern of Hoxc6, Hoxc9, and Hoxc10 in MNs were unaffected, similar to Ezh^MNΔ mice (**Figure 1—figure supplement 1F, H-J**). These observations indicate that depletion of PRC2 function at the time of differentiation does not affect MN class specification or rostrocaudal positional identities.

We next investigated the function of PRC1, which is thought to repress gene expression in a PRC2-dependent manner. We generated mice in which *Rnf2* (also known as *Ring1B*) is conditionally deleted from MN progenitors using *Olig2^Cre* in a global *Ring1* (also known as *Ring1A*) mutant background (Ring1^MNΔ mice) (**Calés et al., 2008**; **del Mar Lorente et al., 2000**). In Ring1^MNΔ mice, Rnf2 (Ring1B) protein was selectively removed from progenitors and post-mitotic MNs (**Figure 1H,I**). The number of Mnx1+ MNs and pattern of *Slc18a3* expression were similar to controls in Ring1^MNΔ mice (**Figure 1J, K and N**). By contrast, Hox proteins normally expressed by forelimb-innervating brachial MNs were not detected, while thoracic and lumbar Hox proteins were ectopically expressed in more rostral MNs (**Figure 1L, M and O–T**, **Figure 1—figure supplement 2C-E**). Brachial-level Hox proteins (Hoxc4, Hoxa5, Hoxc6 and Hoxc8), were selectively depleted from MNs, while thoracic and lumbar *Hox* determinants, (Hoxc9, Hoxc10, and Hoxd10) were derepressed in brachial segments (**Figure 1Q–T**). Brachial MNs also co-expressed Hoxc9 and Hoxc10, which are normally restricted to thoracic and lumbar segments, respectively (**Figure 1—figure supplement 2B**). At thoracic levels, Hoxc9 expression was attenuated and Hoxd10 was ectopically expressed (**Figure 1—figure supplement 2D**). These results show that loss of *Ring1* causes a derepression of caudal *Hox* genes, leading to co-expression of caudal Hox proteins in brachial MNs, without affecting pan-MN molecular features (**Figure 1O**).

### Canonical PRC1 regulates *Hox* expression in MNs

Ring1 can interact with multiple Pcgf proteins, raising the question of which PRC1 configuration contributes to MN patterning. PRC1 containing Pcgf4 is required to establish *Hox* boundaries in MNs (**Golden and Dasen, 2012**), but can exist in two alternative configurations, depending on mutually exclusive incorporation of Cbx (canonical PRC1) or Rybp (variant PRC1) (**Gao et al., 2012**; **Tavares et al., 2012**). We examined the function of PRC1 isoforms first by manipulating Rybp and Cbx expression in MNs. We hypothesized that if canonical PRC1 regulates *Hox* expression then overexpression of Rybp would inhibit binding of Cbx to Ring1, leading to MN phenotypes similar to *Ring1* mutants

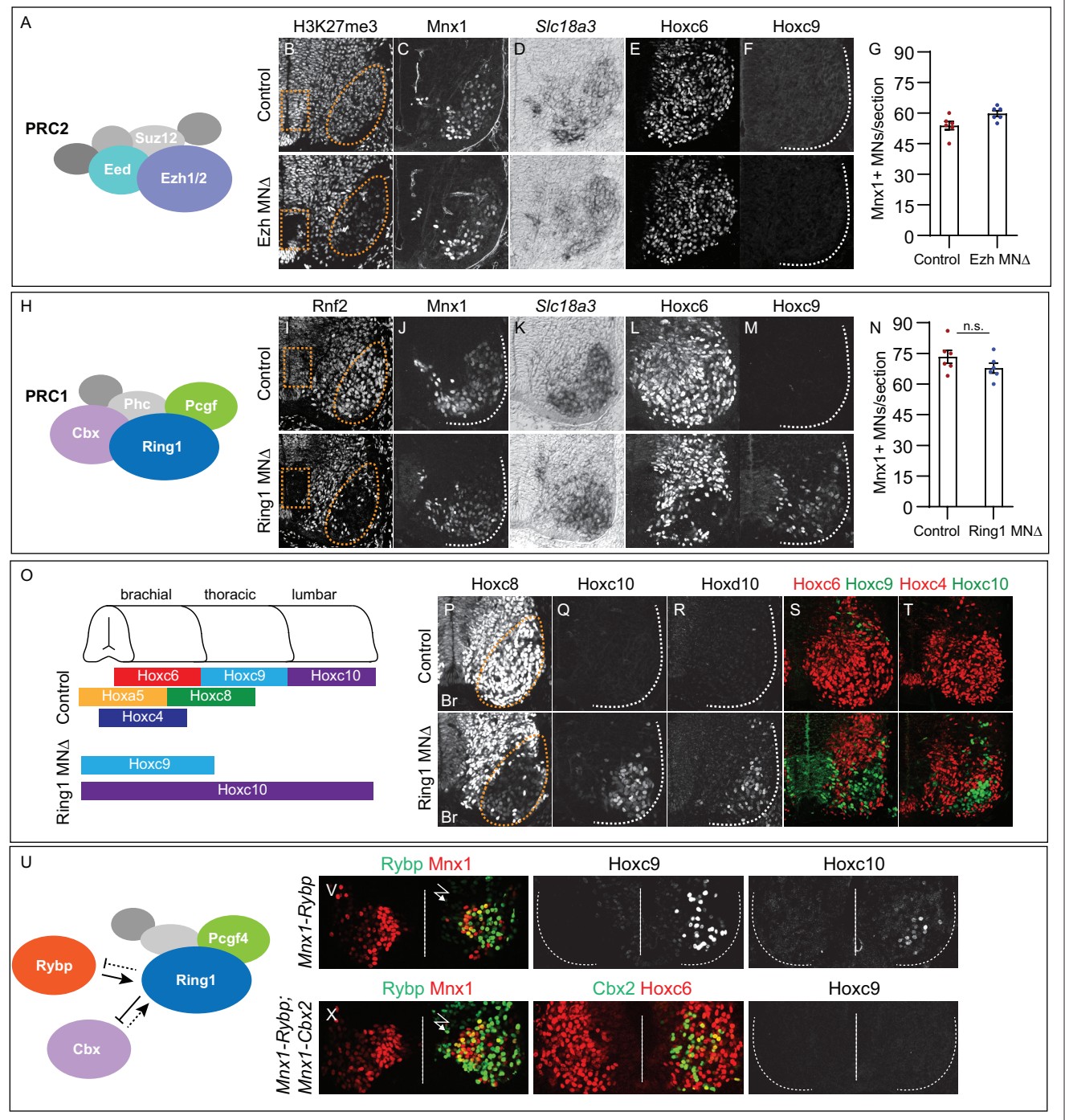

**Figure 1.** Roles of PRC1 and PRC2 in determining of *Hox* expression in spinal MNs. (**A**) Core components of PRC2. (**B**) Brachial spinal sections showing H3K27me3 is depleted from progenitors (boxed region) and post-mitotic MNs (oval) in E11.5 Ezh$^{MNΔ}$ (*Ezh1$^{flox/flox}$::Ezh2$^{flox/flox}$, Olig2$^{Cre}$*) embryos. (**C–D**) MNs express Mnx1 and *Slc18a3 (Vacht)* in Ezh$^{MNΔ}$ mice. (**E–F**) Brachial Hoxc6 expression is normal in Ezh$^{MNΔ}$ mice, and no ectopic Hoxc9 is detected. (**G**) Quantification of MNs: 54±2 Mnx1$^+$ MNs per section in brachial controls, versus 60±1 in Ezh$^{MNΔ}$ mice, n = 6 sections, p = 0.0342, unpaired t-test. (**H**) Core components of PRC1. (**I**) Rnf2 (Ring1B) is selectively removed from progenitors (boxed region) and post-mitotic MNs (oval) in E12.5 Ring1$^{MNΔ}$ (*Ring1$^{-/-}$::Rnf2$^{flox/flox}$, Olig2$^{Cre}$*) mice. (**J–K**) Ring1$^{MNΔ}$ mice express Mnx1 and *Slc18a3*. (**L–M**) Hoxc6 is lost from brachial MNs and Hoxc9 is ectopically expressed in Ring1$^{MNΔ}$ mice. (**N**) Quantification of MNs: 73±3 Mnx1$^+$ MNs in controls, versus 68±2 in Ring1$^{MNΔ}$ mice, n = 6 sections, p = 0.1879, unpaired t-test. (**O**) Summary of changes in MN Hox expression of Ring1$^{MNΔ}$ mice. (**P–R**) Loss of Hoxc8 and ectopic Hoxc10 and Hoxd10 expression in brachial MNs of Ring1$^{MNΔ}$ mice. (**S–T**) Co-labeling of Hoxc6/Hoxc9 and Hoxc4/Hoxc10 in Ring1$^{MNΔ}$ mice, showing ectopically expressed caudal Hox proteins and loss of rostral Hox expression in brachial segments. (**U**) Schematic of Cbx and Rybp interactions in PRC1. (**V**) Misexpression of Rybp in postmitotic MNs under *Mnx1* in chick leads to ectopic Hoxc9 and Hoxc10 expression in brachial MNs. Bolt symbol indicates electroporated side of spinal cord. (**X**) Co-

*Figure 1 continued on next page*

Figure 1 continued

expression of Rybp and Cbx under *Mnx1* fails to induce Hoxc9 in brachial MNs. Panels B-F show brachial sections from E11.5 embryos; I-M, P-T brachial sections from E12.5 embryos; V,X brachial sections of HH st25 chick.

The online version of this article includes the following source data and figure supplement(s) for figure 1:

**Source data 1.** Counts of Mnx1-positive MNs in Ezh and Ring1 mutants.

**Figure supplement 1.** Effects of PRC2 mutations on MN differentiation.

**Figure supplement 1—source data 1.** Quantification of H3K27me3 intensity in MNs of Ezh mutants.

**Figure supplement 2.** Effects of canonical and variant PRC1 mutations on MN differentiation.

(*Figure 1U*). We used chick in ovo neural tube electroporation to express mouse Rybp in postmitotic MNs using the *Mnx1* promoter (*Mnx1-Rybp*). Expression of *Rybp* under *Mnx1* led to ectopic Hoxc9 and Hoxc10 expression at brachial levels (*Figure 1V*). If Rybp acts by displacing Cbx, then elevating Cbx levels should restore normal *Hox* expression. To test this, we co-electroporated *Mnx1-Rybp* and *Mnx1-Cbx2* at equivalent plasmid concentration, and no longer observed ectopic Hoxc9 in brachial segments (*Figure 1X*).

These observations are consistent with canonical PRC1 containing Pcgf4, Cbx, and Ring1 restricting *Hox* expression in MNs. Rybp is expressed by MNs at the time of their differentiation (*Figure 1— figure supplement 2F*), raising the possibility that *Rybp* (or its paralog *Yaf2*) also plays regulatory roles during development. We therefore selectively deleted *Rybp* from MNs using *Olig2^Cre^* mice in a *Yaf2^-/-^* background (Rybp/Yaf2^MNΔ^ mice). Combined deletion of *Rybp* and *Yaf2* did not affect MN generation, Hoxc6, Hoxc9, or Hoxc10 expression (*Figure 1—figure supplement 2F*). These findings indicate that canonical PRC1 maintains appropriate *Hox* expression, while variant PRC1 and PRC2 do not contribute to rostrocaudal patterning at the time of MN differentiation.

## PRC1 is required in MNs for subtype diversification

Hox transcription factors play central roles in establishing neuronal subtype identities through regulating expression of subtype-specific genes. To determine the consequences of altered *Hox* expression in Ring1^MNΔ^ mice, we analyzed the molecular profiles and peripheral innervation pattern of MNs. A key Hox target in MNs is the transcription factor *Foxp1*, which is essential for the differentiation of limb-innervating lateral motor column (LMC) and thoracic preganglionic column (PGC) neurons (*Dasen et al., 2008*; *Rousso et al., 2008*). In Ring1^MNΔ^ mice, Foxp1 expression is lost from brachial and thoracic MNs, and markedly reduced in lumbar MNs (*Figure 2A, B and G*, *Figure 2—figure supplement 1C*). By contrast, in Ezh^MNΔ^, Eed^NEΔ^, and Rybp/Yaf2^MNΔ^ mice, Foxp1 expression was maintained, consistent with the preservation of normal *Hox* profiles in these mutants (*Figure 1—figure supplement 2F*; *Figure 2—figure supplement 1A, B*). In Ring1^MNΔ^ mice, expression of the LMC marker Raldh2 was lost at brachial levels, and the thoracic PGC marker nNos was not detected (*Figure 2C and D*). Moreover, expression of determinants of respiratory phrenic MNs (Scip+ Isl1/2+), was markedly depleted, and likely contributes to the perinatal lethality of Ring1^MNΔ^ mice (*Figure 2—figure supplement 1D*). These results indicate that *Ring1* deletion leads to a loss of genes acting downstream of Hox function in MNs.

We next assessed the impact of *Ring1* deletion on two MN columnar subtypes that are specified independent of Hox function, the hypaxial and median motor columns (HMC and MMC) (*Dasen et al., 2008*; *Jung et al., 2014*). In Ring1^MNΔ^ mice, MMC neurons (Lhx3+ Mnx1+) were generated at normal numbers in thoracic segments, while the number of HMC neurons (Isl1/2+ Mnx1+ Lhx3-) was increased in thoracic and brachial segments (*Figure 2E, F and H*, *Figure 2—figure supplement 1E*,F). The increase in HMC neurons in *Ring1* mutants is likely due to a reversion of presumptive Hox-dependent subtypes to an HMC fate, similar to mice in which Hox function is disrupted (*Dasen et al., 2008*; *Hanley et al., 2016*). We also observed a small population of medial neurons that coexpressed Isl1/2, Mnx1, and Lhx3 (*Figure 2E, F and H*). These cells likely represent undifferentiated MN precursors, but did not appear to express progenitor markers such as Olig2 and Nkx6.1 (*Figure 1—figure supplement 2A*). Mutation in *Ring1* therefore depletes Hox-dependent subtypes, with the remaining MNs having a more ancestral axial or ambiguous subtype identity (*Figure 2P*).

As we observed a dramatic loss of segmentally restricted MN subtypes in Ring1^MNΔ^ mice, we next assessed the impact on peripheral innervation pattern. To trace motor axon projections, we

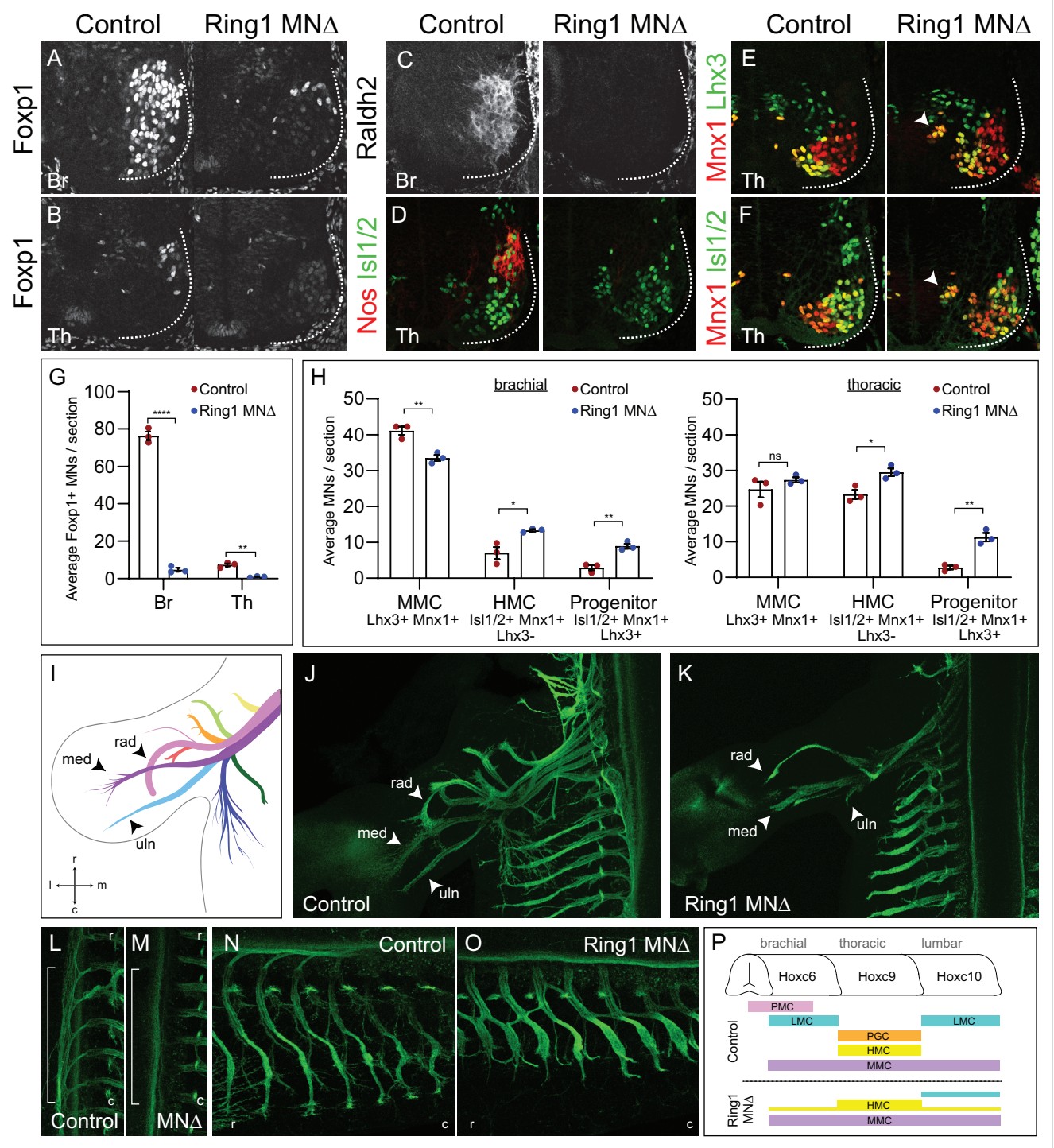

**Figure 2.** *Ring1* is essential for the specification of Hox-dependent MN subtypes. (**A–B**) Foxp1 expression is reduced in brachial (Br) and thoracic (Th) segments of Ring1^MNΔ mice at E12.5. (**C–D**) Expression of the brachial LMC marker Raldh2 and thoracic PGC marker Nos were lost in Ring1^MNΔ mice. (**E–F**) Staining of Mnx1⁺, Lhx3⁺ (MMC) and Mnx1, Isl1/2⁺ (HMC) neurons. In Ring1^MNΔ mice, we also observed a population of medial neurons that coexpressed Isl1/2, Mnx1, and Lhx3 (indicated by arrow heads). (**G**) Quantification of Foxp1 reduction in Br and Th segments. (**H**) Quantification of MMC (Mnx1⁺, Lhx3⁺), HMC (Isl1/2⁺, Mnx1⁺, Lhx3⁻) and medial 'progenitors' (Isl1/2⁺, Mnx1⁺, Lhx3⁺) MNs arrow heads in E,F. Panels G-H show average from n = 3 mice, four sections each animal. Data shown in graphs shown as mean ± SEM. *p < 0.05, **p < 0.01, ****p < 0.0001, unpaired t-test. (**I**) Schematic of nine primary nerves in E12.5 mouse forelimb (Adapted from *Figure 1A Catela et al., 2016*) med = median, rad = radial, uln = ulnar nerves. Rostral (**r**), caudal (**c**), medial (**m**), and lateral (**l**) orientation shown. (**J–K**) Forelimb motor axons of an E12.5 control and Ring1^MNΔ mouse labeled by *Mnx1-GFP*. (**L–M**) Innervation of sympathetic chain ganglia (from PGC neurons) in control and Ring1^MNΔ mice. Bracket shows region of PGC projections along

*Figure 2 continued on next page*

*Figure 2 continued*

rostrocaudal axis. (**N–O**) Innervation of dorsal and ventral axial muscles by MMC and HMC respectively. In Ring1$^{MNΔ}$ mice, HMC motor projections are shorter and thicker. (**P**) Summary of MN columnar organization of control and Ring1$^{MNΔ}$ mice.

The online version of this article includes the following source data and figure supplement(s) for figure 2:

**Source data 1.** Quantification of MN subtypes in Ring1 mutants.

**Figure supplement 1.** Analyses of MN subtypes in PRC mutant mice.

**Figure supplement 1—source data 1.** Quantification of MN subtypes in Ring1 mutants.

---

crossed Ring1$^{MNΔ}$ mice to a *Mnx1-GFP* transgenic reporter, in which all MN axons are labelled with GFP. In control mice, there are nine primary trajectories of forelimb-innervating motor axon at E12.5 (***Figure 2I and J***; ***Catela et al., 2016***). In Ring1$^{MNΔ}$ mice, only three nerve branches, radial, median, and ulnar were visible but appeared prematurely truncated and unbranched (***Figure 2K***). In the trunk, innervation of sympathetic chain ganglia was lost, consistent with a loss of PGC fates (***Figure 2L and M***). Projections to dorsal and ventral axial muscles by MMC and HMC subtypes were maintained in Ring1$^{MNΔ}$ mice, and HMC axons were thicker and shorter than in controls (***Figure 2N and O***, ***Figure 2—figure supplement 1G***). Loss of *Ring1* therefore causes severe defects in innervation pattern, while the trajectories of axial MNs are relatively spared.

## Loss of *Ring1* causes derepression of developmental fate determinants

Polycomb proteins regulate diverse aspects of differentiation by restricting gene expression during development. To investigate changes in gene expression after deletion of *Ring1* genes in an unbiased manner, we performed RNAseq on MNs isolated from Ring1$^{MNΔ}$ mice. We purified MNs from control and Ring1$^{MNΔ}$::*Mnx1-GFP* embryos at E12.5 by flow cytometry (n = 4 *Ring1* mutants, and n = 4 Cre$^-$ controls), and performed RNAseq. Because *Ring1* mutants display segment-specific phenotypes, we collected MNs from brachial, thoracic, and lumbar levels and profiled each population independently.

We identified a total of 1001 upregulated and 641 downregulated genes in Ring1$^{MNΔ}$ mice (log$_2$-FC > 2, FDR < 0.1), with 391 genes upregulated in all three segmental regions, compared to 46 genes that were commonly downregulated (***Figure 3A–C***, ***Figure 3—figure supplement 1A***,C, ***Supplementary file 1***). Thus, many upregulated genes are shared between each segment, while downregulated genes tended to be segment-specific. Strikingly, 76% of the top 100 derepressed genes (by log$_2$-FC) at brachial levels encode transcription factors, and include genes normally involved in the specification of spinal interneurons (e.g. *Gata3*, *Shox2*, *Lhx2*), brain regions (*Foxg1*, *Six6*, *Lhx8*), and non-neuronal lineages (***Figure 3B***).

Although a variety of cell fate determinants were derepressed in Ring1$^{MNΔ}$ mice, our RNAseq analyses provide further evidence that core molecular determinants of MN class identity are preserved in *Ring1* mutants. We found no significant changes in transcription factors (*Isl1*, *Isl2*, *Mnx1*, *Lhx3*, *Lhx4*), guidance molecules (*Slit2*, *Robo1*, *Robo2*, *Dcc*) and neurotransmitter genes (*Chat*, *Acly*) associated with pan-MN features (***Figure 3D***). There was a modest decrease in expression of *Slc18a3* (***Figure 3D***). Expression of genes that mark other excitatory (*Slc17a7*, *Slc17a6*) or inhibitory (*Slc32a1*, *Gad2*, *Gad1*,) neuronal classes were not markedly derepressed in *Ring1* mutants (***Figure 3—figure supplement 1B***). By contrast, expression of genes associated with MN subtype identities was decreased in Ring1$^{MNΔ}$ mice, including genes that mark limb-innervating (e.g. *Foxp1*, *Etv4*, *Raldh2*, *Runx1*), thoracic-specific (*Nos1*, *Cyp26b*), and respiratory (*Alcam*, *Ptn*) subtypes (***Figure 3E***).

To validate these changes in gene expression, we performed mRNA in situ hybridization on a subset of upregulated or downregulated genes. We found that *Gata3*, *Lhx8*, and *Lhx2* were markedly upregulated in MNs of Ring1$^{MNΔ}$ mice (***Figure 4A–F***), while other genes (*Foxg1*, *Six6*, *Pitx2*, *Shox2*, *Nkx2.1*) showed less prominent, but detectable, ectopic expression (***Figure 4—figure supplement 1A-J***). We also examined expression of previously uncharacterized genes that were downregulated in all three segmental regions. We found *Rgs4*, *Uts2b*, *Vat1*, and *Gabra2* were expressed by MNs of control embryos, and were downregulated in *Ring1* mutants (***Figure 4G–L***, ***Figure 4—figure supplement 1K,L***). These results indicate that despite the derepression of multiple fate determinants in Ring1$^{MNΔ}$ mice, core features of MN identity are preserved, but that subtype diversification programs are selectively disrupted.

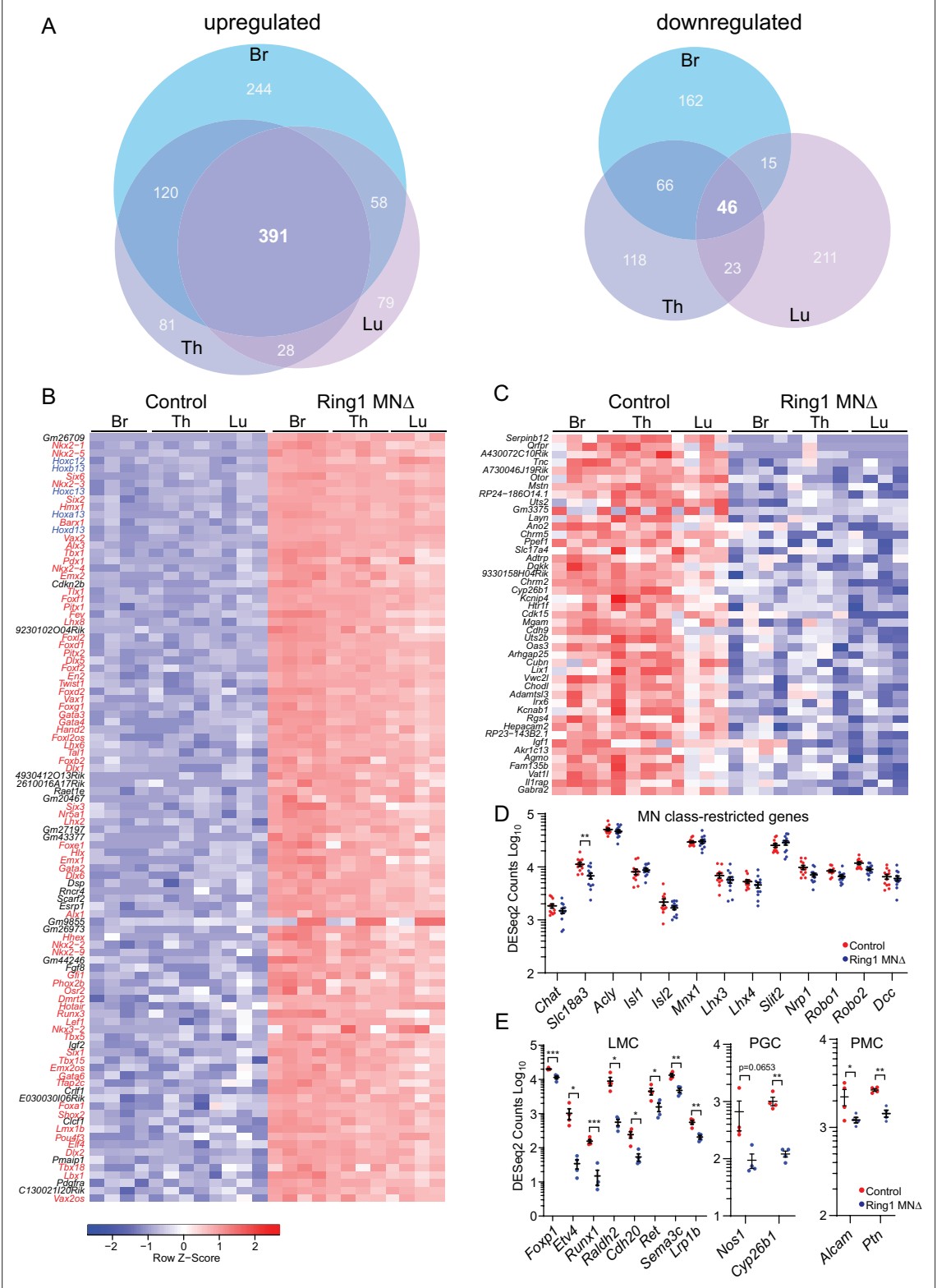

**Figure 3.** *Ring1* is required to restrict transcription factor expression in spinal MNs. (**A**) Venn diagrams showing the number of upregulated (left) and downregulated (right) genes in bracial (Br), thoracic (Th), and lumbar (Lu) MNs upon loss of *Ring1* from MNs by RNAseq (log$_2$-FC > 2, FDR < 0.1). (**B**) Heat map of top 100 upregulated genes (by log2-FC) in Ring1$^{MN\Delta}$ mice and control MNs. Genes shown in blue are *Hox* genes and other transcription factor are shown in red. (**C**) Heat map of 45 genes downregulated in Ring1$^{MN\Delta}$ mice in comparison to control MNs. (**D**) Plots of DESeq2 counts of genes associated with MN class identity. Each data point shows DESeq2 counts for each sample, and segment-specific counts are plotted together. Expression

*Figure 3 continued on next page*

*Figure 3 continued*

of *Slc18a3 (Vacht)* is reduced (padj. = 0.026477) in Ring1$^{MN\Delta}$ mice. (**E**) DESeq2 counts of genes associated with specific MN subtype identities were reduced in Ring1$^{MN\Delta}$ mice. Counts for LMC and PMC markers are from Br segments, PGC from Th segments. Black bars shown in graphs indicate mean ± SEM. *p < 0.05, **p < 0.01, ***p < 0.001, ****p < 0.0001, unpaired t-test.

The online version of this article includes the following figure supplement(s) for figure 3:

**Figure supplement 1.** RNAseq analyses of *Ring1* mutant mice.

## Selective derepression of caudal *Hox* genes in Ring1$^{MN\Delta}$ mice

As our preliminary analyses revealed altered expression in a subset of *Hox* genes in Ring1$^{MN\Delta}$ mice, we further evaluated expression of all 39 *Hox* genes in our RNAseq dataset. Consistent with the analysis of Hox protein expression, rostral *Hox* genes (*Hox4-Hox8* paralogs) were reduced in brachial MNs of Ring1$^{MN\Delta}$ mice, while caudal *Hox* genes (*Hox10-Hox13* paralogs) were derepressed (**Figure 5A**, **Figure 5—figure supplement 1A-C**). We observed derepression of caudal *Hox* genes from each of the four vertebrate *Hox* clusters, including genes not normally detectable in the ventral spinal cord (e.g. *HoxB* genes) (**Dasen et al., 2005**; **Figure 5A**, **Figure 5—figure supplement 1A-C**). The extent of caudal *Hox* derepression correlated with the relative position of genes within a cluster, with *Hox13* paralogs (*Hoxa13, Hoxb13, Hoxc13, Hoxd13*) displaying the most pronounced derepression in Ring1$^{MN\Delta}$ mice (by FC), relative to other caudal *Hox* genes (**Figure 5B**, **Figure 3—figure supplement 1C**). The marked derepression of *Hox13* paralogs was observed in each of the three segmental levels we analyzed, while *Hox10* paralogs (*Hoxa10, Hoxc10*, and *Hoxd10*) were selectively derepressed in brachial and thoracic segments (**Figure 5A**, **Figure 5—figure supplement 1A-C**).

To further validate these findings, we analyzed *Hoxc13* and *Hoxb13* expression by in situ hybridization. In controls *Hoxc13* is restricted to sacral segments, while *Hoxb13* is not detected in MNs (**Figure 5C**). In Ring1$^{MN\Delta}$ mice, both *Hoxb13* and *Hoxc13*, were de-repressed in MNs throughout the rostrocaudal axis (**Figure 5C**). In addition, in situ hybridization of *Hoxc6* and *Hoxc9* expression revealed reduced expression at brachial and thoracic levels, respectively (**Figure 5—figure supplement 1D**). By contrast, ectopic expression of *Hoxb13* and *Hoxc13* was not observed in Ezh$^{MN\Delta}$ mice (**Figure 5—figure supplement 2A**,B). Thus, ectopic expression of *Hox13* paralogs in *Ring1* mutants is associated with a loss of rostral *Hox* gene expression.

## *Ring1* is essential to maintain MN chromatin topology

Our findings indicate that in the absence of *Ring1* genes, a broad variety of cell fate determinants are ectopically expressed in MNs, while only a subset of caudal *Hox* genes are derepressed. As PRC1 restricts gene expression through chromatin compaction, we investigated whether removal of *Ring1* leads to changes in DNA accessibility at derepressed loci. We used Assay for Transposase-Accessible Chromatin with high-throughput sequencing (ATACseq) to identify genomic regions which have gained or lost accessibility in Ring1$^{MN\Delta}$ mice. We purified *Mnx1-GFP* MNs at E12.5 from control and Ring1$^{MN\Delta}$ embryos at brachial, thoracic, and lumbar levels, and performed ATACseq. In control samples, we observed a progressive opening of caudal *Hox* genes in more rostral segments. For example, the accessibility of *Hoxc9* and *Hoxa9* increases from brachial to thoracic segments, while *Hoxc10, Hoxc11*, and *Hoxa10* are more accessible in lumbar segments (**Figure 6—figure supplement 1A**, B).

To determine which genomic regions gained accessibility in *Ring1* mutants, we compared ATACseq profiles between MNs of control and Ring1$^{MN\Delta}$ mice. We identified a total of 2305 loci that gained accessibility in Ring1$^{MN\Delta}$ embryos, 324 (14%) of which were common to all three segmental levels (**Figure 6A**, **Supplementary file 2**). Common genes included caudal *Hox* genes (*Hoxa13, Hoxc13*) and other transcription factors (e.g. *Foxg1, Lhx2, Pitx2*) that were derepressed in our RNAseq analyses (**Figure 6C**). We also identified 1264 loci that lost accessibility in Ring1$^{MN\Delta}$ mice, 39 (3%) of which were common to all three segments (**Figure 6A**). Thus, similar to our RNAseq results, loss of *Ring1* leads to increased accessibility in many genes that are shared among each segment, while genes that lose accessibility tend to be segment-specific.

We next examined the overlap between transcripts that were upregulated and loci that gained accessibility in Ring1$^{MN\Delta}$ mice. We found that 18% (148/813) of genes that were ectopically expressed at brachial segments also gained accessibility in Ring1$^{MN\Delta}$ mice (**Figure 6B**). We found 19 genes, including *Hoxa13* and *Hoxc13*, were derepressed and gained accessibly in all three segments

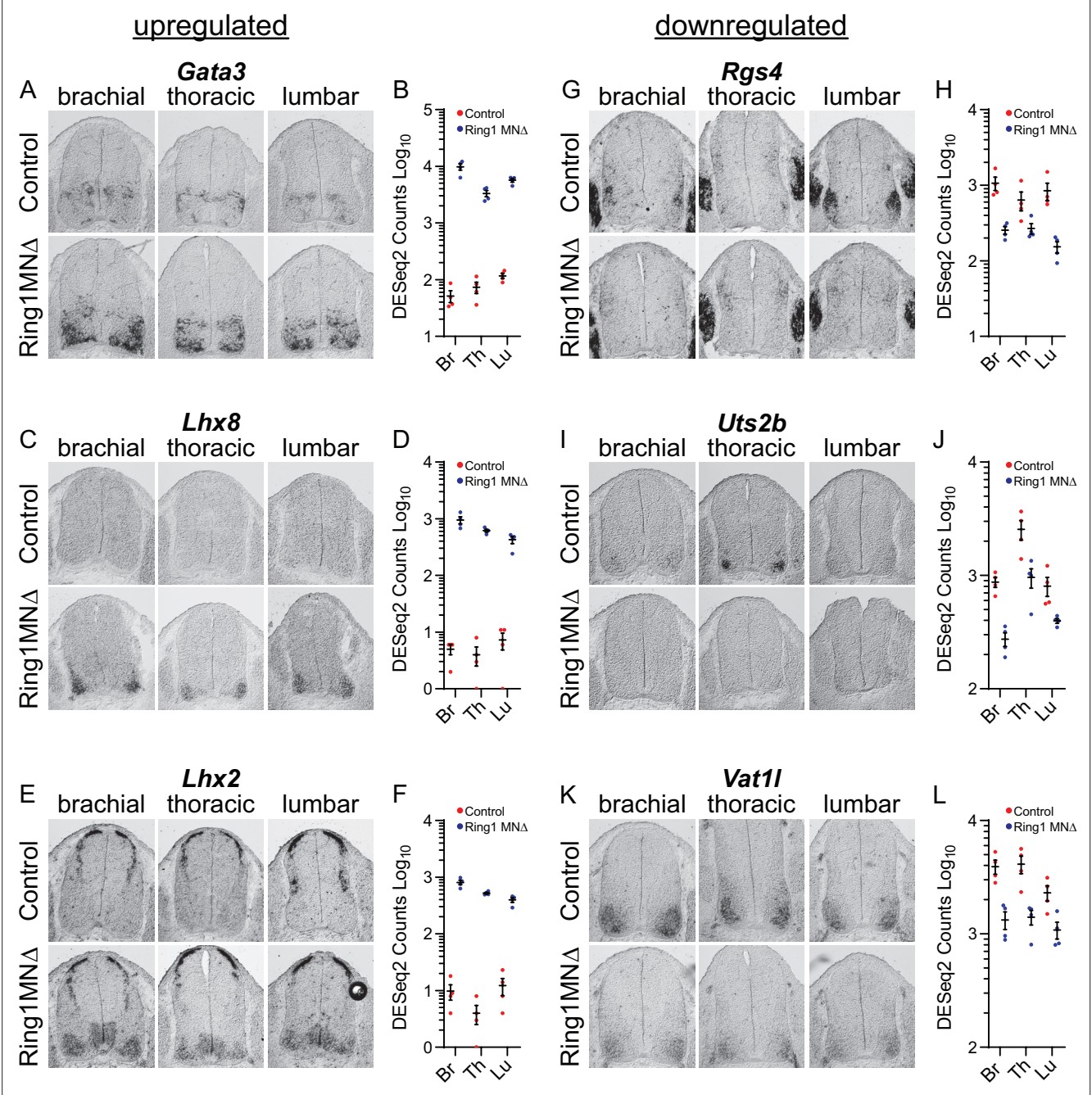

**Figure 4.** Analyses of misregulated genes in *Ring1* mutants. (**A,C,E**) In situ mRNA hybridization of selected upregulated genes from Ring1$^{MN\Delta}$ RNAseq. Images show sections of brachial, thoracic, and lumbar segments from E12.5 control and Ring1$^{MN\Delta}$ mice. (**B,D,F**) Graphs of DESeq2 counts for upregulated gene in each segment. Data points show DESeq2 counts from MNs of individual animals from indicated segments. (**G, I, K**) Analyses of downregulated genes by in situ hybridization. *Rgs4* and *Uts2b* displayed elevated expression in specific segmental levels in controls, suggesting that a subset of the commonly downregulated genes are also Hox-dependent. *Rgs4* expression is normally elevated in LMC neurons (panel G), while *Uts2b* is elevated in thoracic segments of controls (panel I) (**H,J,L**) Graphs of DESeq2 counts for each downregulated gene in each segment.

The online version of this article includes the following figure supplement(s) for figure 4:

**Figure supplement 1.** Validation of misregulated genes from Ring1$^{MN\Delta}$ RNAseq.

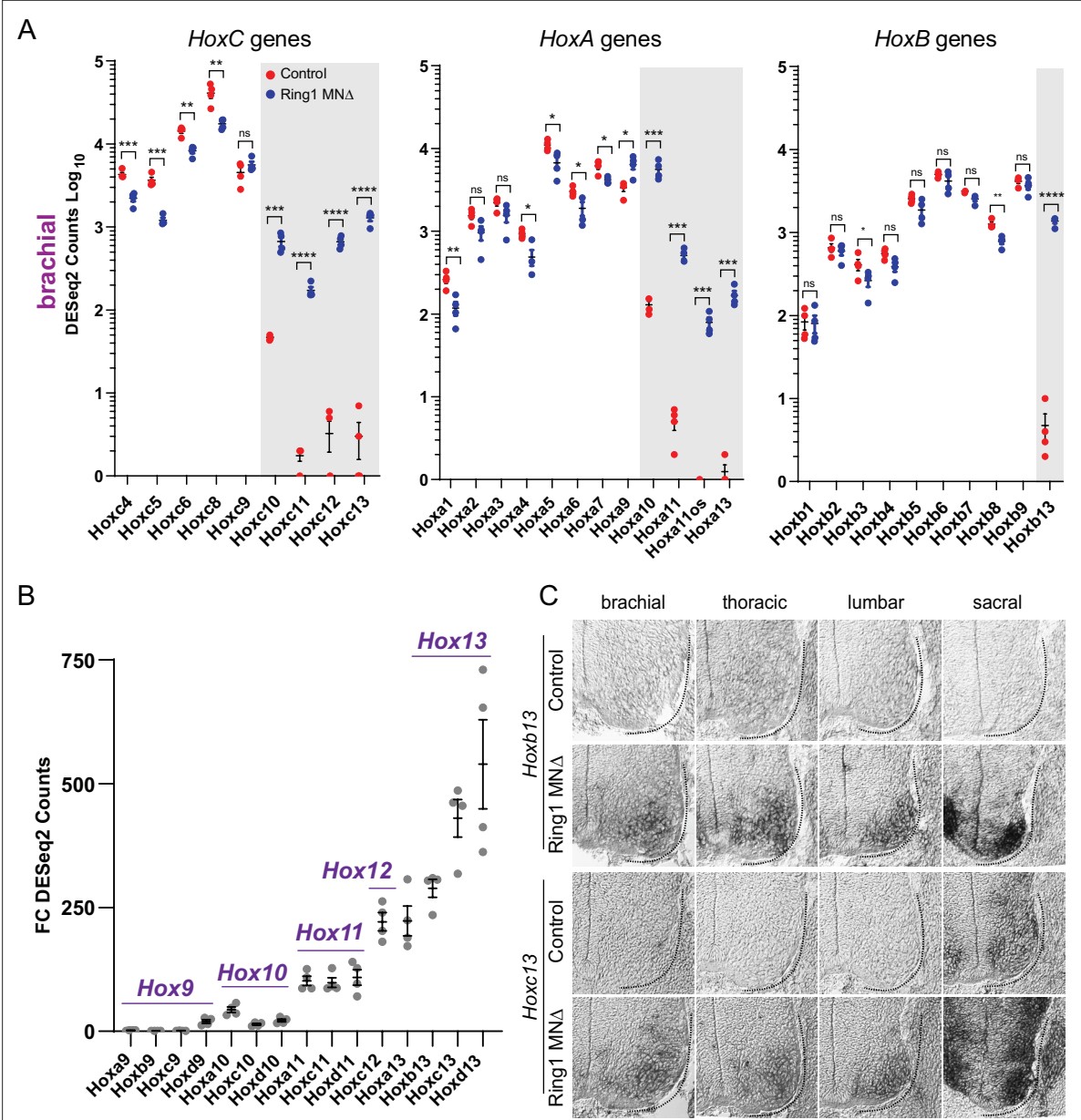

**Figure 5.** Derepression of caudal *Hox* genes in *Ring1* mutants. (**A**) DESeq2 counts of *HoxC*, *HoxA,* and *HoxB* cluster genes in brachial segments in control and Ring1$^{MNΔ}$ mice showing derepression of caudal *Hox* genes. Gray shaded regions highlight *Hox* genes that are derepressed in Ring1$^{MNΔ}$ mice. *Hoxc9* does not show significant derepression, likely because it is normally expressed by caudal brachial MNs. Black bars shown in graphs indicate mean ± SEM. *p < 0.05, **p < 0.01, ***p < 0.001, ****p < 0.0001, unpaired t-test. (**B**) Comparison of *Hox9-Hox13* paralog gene derepression in brachial segments. Graph shows absolute fold changes of DESeq2 counts, showing a marked increase for caudal *Hox13* paralogs in Ring1$^{MNΔ}$ mice. Each data point shows individual counts for *Ring1* mutants/average of controls. (**C**) In situ of *Hoxb13* and *Hoxc13* mRNA transcripts in E12.5 embryos. *Hoxb13* is normally not detectable in spinal cord, but is derepressed in MNs in Ring1$^{MNΔ}$ mice. *Hoxc13* transcripts are normally restricted to sacral segments but derepressed in rostral segments in Ring1$^{MNΔ}$ mice.

The online version of this article includes the following figure supplement(s) for figure 5:

**Figure supplement 1.** Derepression of caudal *Hox* genes in Ring1$^{MNΔ}$ mice.

**Figure supplement 2.** Expression of *Hoxc13* and *Hoxb13* in control and *Ezh* mutant mice.

(*Figure 6—figure supplement 1C*). Since lumbar segments still retain features of Hox-dependent subtypes, we also compared the overlap between genes that were upregulated and gained accessibility in brachial and thoracic MNs. We identified 73 genes, 56 (77%) of which encode transcription factors, including each of the caudal *Hox* genes we found by RNAseq (*Figure 6B*, *Figure 6—figure*

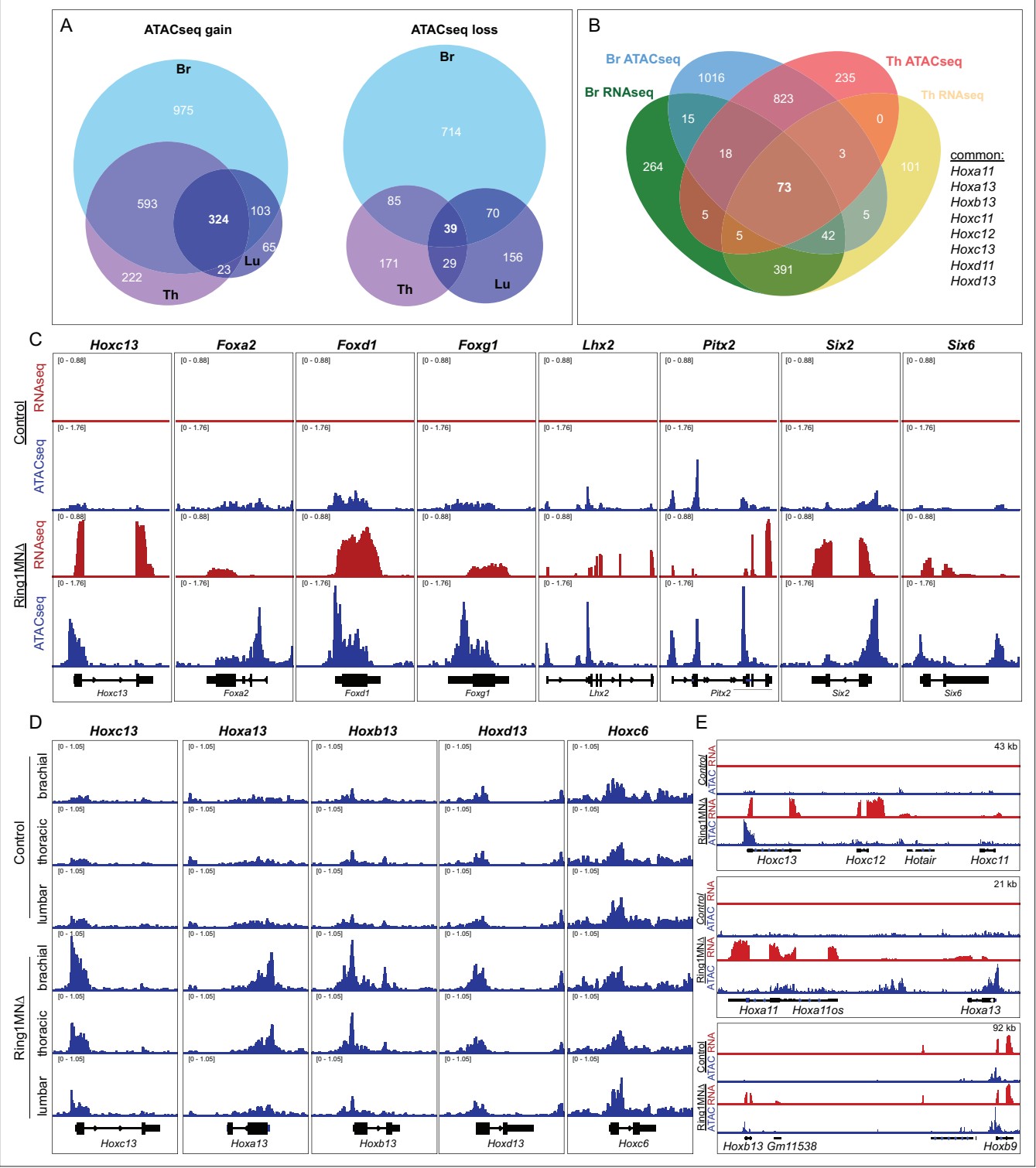

**Figure 6.** *Ring1* is essential for maintaining chromatin topology at cell fate determining genes. (**A**) Proportional Venn diagram showing the number of genes that gain (left) or lose (right) chromatin accessibility in brachial, thoracic and lumbar segments in Ring1^MNΔ mice. (**B**) Venn diagram of upregulated genes from RNAseq and genes that gained accessibility in ATACseq of brachial and thoracic segments. (**C**) IGV browser views of selected genes that are upregulated and gained chromatin accessibility in controls and Ring1^MNΔ mice in brachial segments. (**D**) IGV browser views of ATACseq tracks in controls and Ring1^MNΔ mice in brachial, thoracic, and lumbar segments. *Hox13* paralogs gain chromatin accessibility in each segment, while reduced brachial *Hoxc6* expression is not associated with a loss of chromatin accessibility. (**E**) Comparison of IGV browser views of RNAseq and ATACseq tracks of caudal *HoxC, HoxA,* and *HoxB* genes in brachial segments.

*Figure 6 continued on next page*

Figure 6 continued

The online version of this article includes the following figure supplement(s) for figure 6:

**Figure supplement 1.** Integrative analysis of RNAseq and ATACseq of MNs in Ring1^MNΔ mice.

*supplement 1C*). The gain of accessibility at transcription factor-encoding genes was prominent near transcription start sites (*Figure 6C*), and regions that gained accessibility in Ring1^MNΔ neurons correspond to regions shown to be bound by Rnf2 (Ring1B) in ES cells (*Figure 6—figure supplement 1D*; *Bonev et al., 2017*).

In agreement with our RNAseq analysis, *Hox13* paralogs showed pronounced increases in accessibility, with each of the four *Hox13* paralogs gaining accessibility in each of the three segmental levels (*Figure 6D and E*). By contrast, rostral *Hox4-Hox8* paralogs, which are transcriptionally downregulated in *Ring1* mutants, were not among that targets that lost accessibility (*Figure 6D*). These results suggest that reduced expression of rostral *Hox4-Hox8* paralogs in *Ring1* mutants is not due to a loss in chromatin accessibility.

## Hox13 paralogs repress rostral *Hox* genes by engaging accessible chromatin domains

As PRC1-mediated chromatin compaction is essential for *Hox* repression, it is surprising that *Hox4-Hox8* paralogs are diminished in *Ring1* mutants, without a noticeable reduction in chromatin accessibility at these loci. Because cross-repressive interactions between *Hox* genes themselves are an important regulatory mechanism determining *Hox* boundaries (*Philippidou and Dasen, 2013*), it is possible that ectopically expressed *Hox13* paralogs directly repress multiple *Hox* genes in Ring1^MNΔ mice. To test this, we first analyzed MNs derived from ES cells (ESC-MNs) in which *Hoxc13* expression can be induced upon doxycycline (Dox) treatment (*Bulajić et al., 2020*). ESC-MNs differentiated via RA and Sonic Hedgehog agonist are characterized by expression of anterior *Hox* paralogs. RNAseq analyses revealed that *Hoxa4*, *Hoxa5*, *Hoxc4*, and *Hoxc5* expression were markedly reduced after Dox-induced *Hoxc13* expression in MN progenitors, compared to non-induced ESC-MNs (*Figure 7A*, *Figure 7—figure supplement 1A-C*). This repressive effect also appears to be direct, as ChIPseq analysis indicates that Hoxc13 can bind at multiple *Hox* genes (*Figure 7A*).

Hoxc13 and Hoxa13 can target inaccessible chromatin (*Bulajić et al., 2020*; *Desanlis et al., 2020*), while repression by the *Drosophila* Hox protein Ubx is associated with chromatin compaction (*Loker et al., 2021*). To examine the possible effects of caudal Hox proteins on local chromatin structure, we compared accessibility at *Hox* loci in both control and *Ring1* mutants, relative to the location of Hoxc13 binding sites. At *Hoxc4, Hoxc5, Hoxa4,* and *Hoxa5*, Hoxc13 bound sites correspond to regions that are accessible in both control and *Ring1* mutant MNs (*Figure 7A*, *Figure 7—figure supplement 1A-C*). Hox13 proteins therefore appear to repress rostral *Hox* genes, in part, through binding at pre-existing accessible regions.

To test whether *Hox13* paralogs can repress multiple *Hox* paralogs in vivo, we used chick neural tube electroporation to express *Hoxa13 and Hoxb13* within brachial, thoracic, and lumbar segments, and assessed Hox protein expression. We electroporated *pCAGGs-Hoxa13-ires-nucGFP* or *pCAGGs-Hoxb13-ires-nucGFP* and found that both cell-autonomously repressed expression of brachial, thoracic, and lumbar Hox proteins (Hoxc4, Hoxa5, Hoxc6, Hoxc8, Hoxc9, and Hoxc10) (*Figure 7B–D*). By contrast, misexpression of *Hoxa13* and *Hoxb13* did not affect expression of the general MN markers Isl1/2 and Mnx1 (*Figure 7B–D*). These results indicate that Hox13 paralogs can repress multiple *Hox* genes, and likely contribute to the MN fate specification defects of Ring1^MNΔ mice.

## Discussion

The Polycomb group encompass a large and diverse family of proteins essential for maintaining epigenetic memory of early patterning events. Classically, PcG-mediated repression is thought to depend on recruitment of PRC1 through recognition of histone methylation marks deposited by PRC2 activity. Although alternative, H3K27me3-independent, mechanisms of PRC1 repression have been described (*Tavares et al., 2012*), the relative contribution of PRC1 and PRC2 to neuronal fate specification have not been resolved. In this study, we found that genetic depletion of PRC2 components

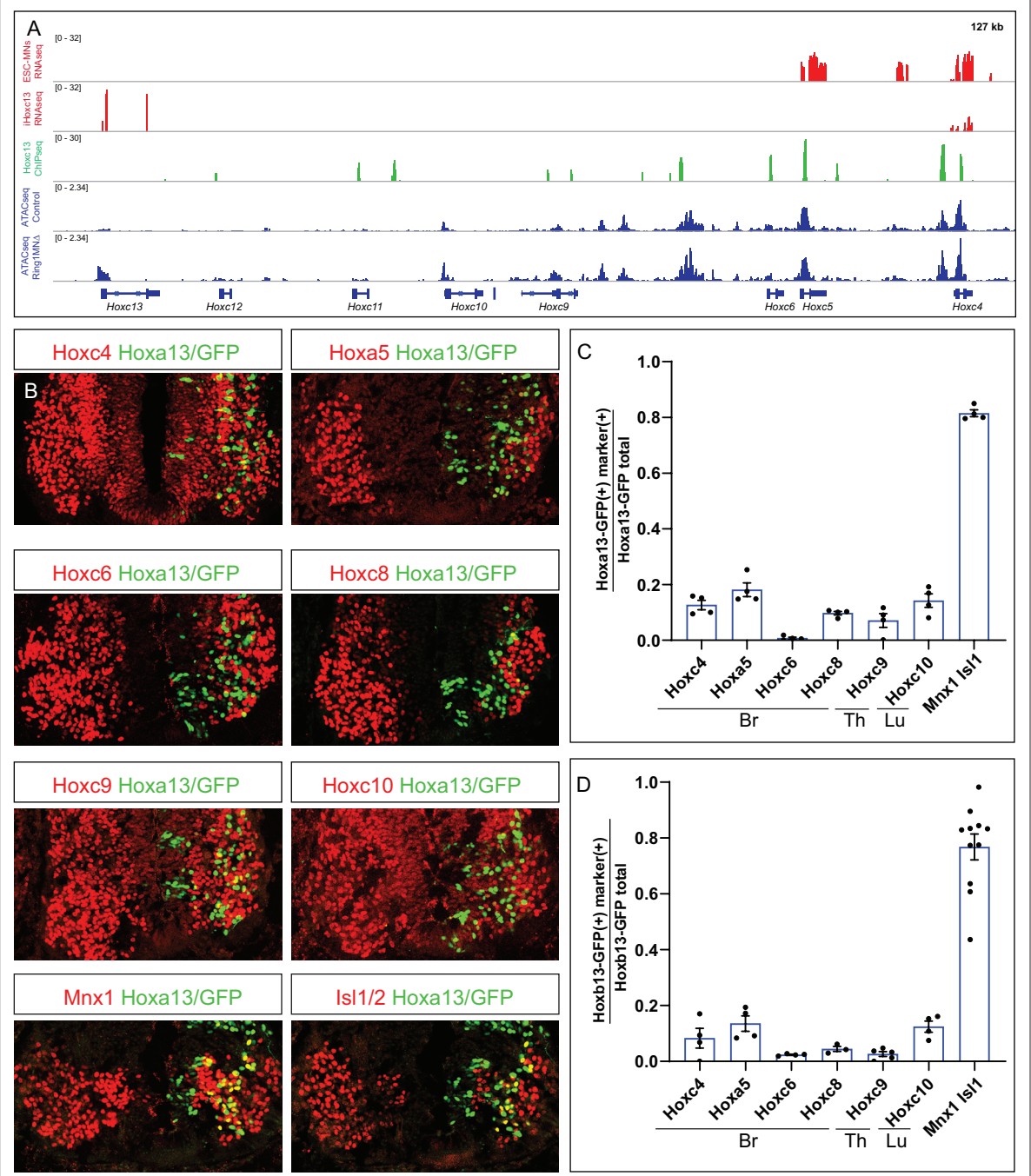

**Figure 7.** Hoxa13 and Hoxb13 repress multiple *Hox4-Hox10* paralogs. (**A**) Effects of *Hoxc13* induction in ESC-MNs. Top panels show IGV browser views (log scale) of RNAseq (red) and ChIPseq tracks (green) at the *HoxC* cluster. Induced Hoxc13 binds near multiple *HoxC* genes and represses expression of *Hoxc4* and *Hoxc5*. Bottom panels show ATACseq in brachial control and *Ring1* mutant MNs at the *HoxC* cluster (blue). (**B**) Misexpression of *Hoxa13-ires-nucGFP* represses more rostral *Hox* genes. Hoxc4, Hoxa5, Hoxc6, and Hoxc8 were analyzed in brachial segments, Hoxc9 in thoracic segments, and Hoxc10 in lumbar segments. Mnx1 and Isl1 images are from brachial segments. (**C,D**) Quantification of percentage of electroporated cells which retained the expression of indicated Hox protein and MN markers upon *Hoxa13* or *Hoxb13* misexpression. Data in graphs show percentages of GFP+ neurons that express the markers indicated on x-axis. Data are from at least four embryos, four sections each embryo, and show mean ± SEM. Quantified electroporated cells were selected from the ventrolateral spinal cord, where MNs reside. Mnx1 and Isl1/2 quantification shows percentage of GFP+ neurons that express either marker, and taken from Br segments in panel C, and Br and Th segments in D.

The online version of this article includes the following source data and figure supplement(s) for figure 7:

**Source data 1.** Quantification of Hoxa13- and Hoxb13-electroporated embryos.

**Figure supplement 1.** Comparison of Hoxc13 binding and accessibility in control and Ring1^MNΔ MNs.

has no discernable impact on neural class or subtype diversification programs at the time of MN differentiation. By contrast, a core PRC1 subunit, Ring1, is required to restrict transcription factor expression in the CNS, maintain neuronal subtype-specific chromatin topology, and determine rostrocaudal boundaries of *Hox* expression. Our findings indicate PRC1 plays a key role in regulating expression of cell fate determinants during MN differentiation, safeguarding neurons from acquiring inappropriate gene regulatory programs.

## PRC functions in Neural development

Polycomb proteins function in diverse aspects of CNS maturation, including the temporal transition from neurogenesis to gliogenesis, maintenance of adult neural stem cell fates, and restriction of gene expression in mature neurons (*Di Meglio et al., 2013*; *Fasano et al., 2009*; *Hirabayashi et al., 2009*; *Molofsky et al., 2003*; *von Schimmelmann et al., 2016*). The role of PRCs in neural subtype diversification has been challenging to study, in part, due the complexity and dynamic temporal regulation of its constituents. By selectively removing core PRC1 and PRC2 subunits from neural progenitors, we determined the relative contributions of these chromatin-associated complexes to neuronal diversification. We found that PRC1 represses a broad variety of cell fate determinants, with the majority of highly derepressed genes in *Ring1* mutants encoding transcription factors. Ectopic expression of Ring1-regulated targets is associated with a disruption in chromatin topology, resulting in a gain in accessibility of derepressed genes.

Despite the derepression of multiple fate determinants in the absence of PRC1 function, markers of MN class identity (e.g. Mnx1, Isl1/2, and Lhx3) are largely unchanged, and MNs do not appear to acquire features of alternate neuronal class fates, beyond expression of fate-defining transcription factors. This observation suggests that once a class identity has been established, ectopic expression of fate determinants is insufficient to respecify basic neuronal features, such as neurotransmitter identity. In *C. elegans* the ability of transcription factors to reprogram neuronal class identity requires removal of histone deacetylase proteins (*Tursun et al., 2011*), suggesting that additional regulatory constraints, such as histone acetylation and/or DNA-methylation, restricts the ability of ectopic transcription factors to interfere with class-specific programs in *Ring1* mutants.

While basic elements of MN fate are maintained in the absence of PRC1, their differentiation into molecularly distinct subtypes is severely compromised. The loss of subtype-specific programs in brachial and thoracic segments can be attributed to the attenuation of expression of rostral *HoxA* and *HoxC* cluster paralogs, as mice lacking these *Hox* genes show a similar loss in MN subtype features (*Jung et al., 2010*; *Jung et al., 2014*; *Philippidou et al., 2012*). Selective loss of *Hox* function in MNs also appears to account for the observation that the majority of downregulated genes in *Ring1* mutants are segment-specific.

Our results indicate that in *Ring1* mutants brachial MNs ectopically express *Hox9-Hox13* paralogs, while expression of *Hox* genes normally expressed by brachial MNs is diminished. As a consequence, MNs fail to acquire a limb-innervating LMC molecular fate, and express markers indicative of MMC and HMC subtypes. These observations are consistent with a model in which the Hox code of MNs is scrambled in *Ring1* mutants, reverting to the fate of ancestral axial subtypes. However, we cannot formally rule out the possibility that some MNs acquire a sacral Hox13$^+$ fate, although we did not observe ectopic expression of markers of sacral PGC identity (e.g. nNos, Foxp1).

We previously found that acute depletion of the canonical PRC1 component Pcgf4 (Bmi1) in chick leads to ectopic *Hoxc9* expression in brachial segments and a switch of LMC neurons to a thoracic PGC fate (*Golden and Dasen, 2012*). By contrast in *Ring1* mutants, multiple caudal *Hox* genes (*Hox9-Hox13* paralogs) are derepressed, and the remaining MNs have an axial subtype fate. While the exact mechanisms of these difference are unclear, they could reflect differences in the timing and spatial extent of the manipulations. In *Ring1* mutants PRC1 function is depleted from MN progenitors throughout the rostrocaudal axis, whereas *Bmi1* knockdown was performed in brachial segments just prior to differentiation.

## PRC-dependent and independent mechanisms of rostrocaudal patterning in the CNS

Our results show that mutation in *Ring1* genes leads to ectopic expression and increased accessibility of many transcription factor-encoding genes that are common to each segment, while genes that are

downregulated and lose accessibility tend to be segment-specific. These observations indicate that PRC1 is required to restrict expression of multiple cell-fate determinants throughout the rostrocaudal axis. By contrast, the downregulation of segment-specific genes can be attributed to the loss of Hox-dependent MN columnar subtypes, due to altered *Hox* gene expression in PRC1 mutants.

We found that removal of PRC1 leads to the derepression and increased accessibility of caudal *Hox* genes. Loss of PRC1 does not result in a sustained derepression of all *Hox* genes, as *Hox* genes normally expressed by brachial- and thoracic-level MNs are attenuated in the absence of *Ring1*. These findings are reminiscent of the function of PRCs in *Drosophila*, where loss of PRC function leads to ectopic expression of the caudal *Hox* gene *Abd-b* and repression of *Ubx* expression (***Struhl and Akam, 1985***; ***Struhl and White, 1985***), likely reflecting cross-repressive interactions between caudal Hox proteins and rostral *Hox* genes. Notably, reduced rostral *Hox* gene expression in *Ring1* mutants is not a reflection of reduced chromatin accessibility at these targets, consistent with a model in which this form of repression is not associated with global changes in chromatin structure.

In *Ring1* mutants, *Hox13* paralogs are derepressed in MNs, and appear to contribute to the loss of MN subtypes. Derepression of *Hox13* paralogs alone does not appear to account for all of the specification defects in *Ring1* mutants, as *Hox10* paralogs (*Hoxa10*, *Hoxc10*, and *Hoxd10*) were expressed normally in lumbar segments. Hoxc9 can also repress *Hox4-Hox8* paralogs in MNs through direct binding (***Jung et al., 2010***), and likely contributes to the *Ring1* mutant phenotype. Derepression of multiple caudal *Hox* paralogs therefore appears to have a cumulative effect on repressing brachial and thoracic *Hox* genes. In conjunction with studies in *Drosophila*, these finding indicate a conserved mechanism of rostrocaudal patterning, in which PRCs establish rostral *Hox* boundaries, while Hox cross-repressive interactions define caudal boundaries.

## PRC1 functions in maintaining neuronal subtype fate

Studies of neuronal and non-neuronal development provide compelling evidence that PRC2 activity is required to restrict gene expression in early development (***Ezhkova et al., 2009***; ***Gentile et al., 2019***; ***Snitow et al., 2015***; ***Yaghmaeian Salmani et al., 2018***). Our studies show that *Hox* boundaries are unaffected after genetic removal of PRC2 function, which leads to diminished H3K27me3 by the time of MN differentiation. Mutation of *Ezh* genes or *Eed* depletes H3K27me3 from progenitors and postmitotic neurons by E11.5, without appreciably affecting MN generation, *Hox* expression, or downstream Hox effectors. As the axial identity of progenitor cells appears to be shaped prior to neurogenesis (***Metzis et al., 2018***), our PRC2 manipulations likely remove H3K27me3 after Polycomb repression has been initiated. Consistent with this idea, our analyses of *Ezh* mutants as well as previous analyses of Eed^NEΔ mice (***Yaghmaeian Salmani et al., 2018***), showed detectable levels of H3K27me3 in the MN domain up until E10.5, indicating that some H3K27me3 remains during the phase in which anterior *Hox* boundaries are established.

How does PRC1 restrict expression of fate determinants in the absence of PRC2 activity? One plausible scenario is that residual H3K27me3 is transmitted through cell division in PRC2 mutants and is sufficient to recruit canonical PRC1 and maintain target gene repression. As cells divide, newly synthesized histones are presumably devoid of H3K27me3 in PRC2 mutants, leading to replication-coupled dilution of H3K27me3. During neural differentiation, the rate of cell division decreases (***Kicheva et al., 2014***; ***Wilcock et al., 2007***), potentially limiting H3K27me3 depletion, and enabling PRC1 to bind at target loci, even in the absence of de novo H3K27 methylation. Consistent with this idea, recent studies on the effects of PRC2 depletion in intestinal stem cells indicate that ~40% of residual H3K27me3 can maintain PRC repression (***Jadhav et al., 2020***), although the extent of derepression varies with gene target. Although we observed no MN phenotypes in *Ezh* or *Eed* mutants, later-born oligodendrocytes derived from the MN progenitor domain have been shown to depend on PRC2 function for their differentiation (***Wang et al., 2020***). This more pronounced effect on glial development likely reflects further dilution of H3K27me3 through cell division in PRC2 mutants.

Stabilization of PRC1 at repressed loci could also maintain target repression independently of de novo H3K27 methylation. Repression by canonical PRC1 has been shown to depend on the formation of phase-separated subnuclear structures (Polycomb bodies) assembled through polymerization of Polyhomeotic-like (Phc) and/or Cbx proteins (***Isono et al., 2013***; ***Plys et al., 2019***; ***Tsuboi et al., 2018***). Mutation of *Phc2* in mice leads to ectopic expression of *Hox* genes, with *Hoxb13* among the most robustly derepressed targets (***Isono et al., 2013***). One possibility is that PcG-mediated

repression may not exclusively depend on anchoring of PRC1 through H3K27me3, but is also maintained through Phc-mediated chromatin compaction at specific loci. Alternatively, PRC2-independent repression by PRC1 could be facilitated through its H2A-ubiquitination activity, as has been suggested in other systems (*Tsuboi et al., 2018*).

We suggest that in the early phases of development PRC2 activity defines the sites of maintained H3K27 tri-methylation, initiates PRC1 recruitment, and restricts transcription factor expression in MNs. These activities are likely initiated prior to MN progenitor specification, as recent studies have demonstrated that the establishment of rostrocaudal positional identities can be specified before neural induction (*Metzis et al., 2018*). Patterning morphogens, such as RA and FGF, may act on stem cells to establish the pattern of H3K27me3 at *Hox* loci prior to neurogenesis. This early rostrocaudal patterning step may ultimately serve to coordinate *Hox* expression in the neural tube and surrounding mesodermally-derived tissues. The subsequent switch to reliance on PRC1 function could reflect a general mechanism of gene regulation in neuronal subtypes that become terminally differentiated during development.

# Materials and methods

### Key resources table

| Reagent type (species) or resource | Designation | Source or reference | Identifiers | Additional information |
|---|---|---|---|---|
| Genetic reagent (*M. musculus*) | *Ezh1*flox/flox | PMID:23122289 | MGI:1097695 | |
| Genetic reagent (*M. musculus*) | *Ezh2*flox/flox | PMID:12496962 | MGI:107,940 | |
| Genetic reagent (*M. musculus*) | *Olig2*Cre | PMID:18046410 | MGI: 3774124 | |
| Genetic reagent (*M. musculus*) | *Ring1*-/-::*Rnf2*flox/flox | PMID:18039844, 11060235 | MGI:1101770, MGI:1101759 | |
| Genetic reagent (*M. musculus*) | *Yaf2*-/-::*Rybp*flox/flox | PMID:27705745 | MGI:1914307 MGI:1929059 | |
| Genetic reagent (*M. musculus*) | *Mnx1-GFP* | PMID:10482234 | | |
| Biological sample (chicken eggs) | SPF Eggs | Charles River | 10100332 | |
| Antibody | Anti-Hoxc4 (Rabbit polyclonal) | PMID:16269338 | | (1:16,000) |
| Antibody | Anti-Hoxa5 (Rabbit polyclonal) | PMID:16269338 | | (1:16,000) |
| Antibody | Anti-Hoxc6 (Guinea pig polyclonal) | PMID:11754833 | RRID: AB_2665443 | (1:16,000) |
| Antibody | Anti-Hoxc6 (Rabbit polyclonal) | Aviva Systems Biology | Cat# ARP38484; RRID: AB_10866814 | (1:32,000) |
| Antibody | Anti-Hoxc8 (Mouse monoclonal) | Covance | RRID: AB_2028778 | (1:4000) |
| Antibody | Anti-Foxp1 (Rabbit polyclonal) | PMID:18662545 | RRID: AB_2631297 | (1:32,000) |
| Antibody | Anti-Isl1/2 (Guinea pig polyclonal) | Jessell lab | | (1:10,000) |
| Antibody | Anti-Rnf2 (Goat polyclonal) | Abcam | Cat# ab3832, RRID: AB_304100 | (1:2000) |
| Antibody | Anti-Rnf2 (Rabbit polyclonal) | Abcam | Cat# ab101273, RRID: AB_10711495 | (1:5000) |
| Antibody | Anti-Rybp (Rabbit monoclonal) | Abcam | Cat# ab185971 | (1:2000) |
| Antibody | Anti-Cbx2 (Rabbit polyclonal) | Bethyl | Cat# A302-524A, RRID:AB_1998943 | (1:5000) |
| Antibody | Anti-H3K27me3 (Rabbit polyclonal) | Cell Signaling | Cat# 9733, RRID: AB_2616029 | (1:2000) |
| Antibody | Anti-Hoxc9 (Guinea pig polyclonal) | PMID:20826310 | RRID:AB_2636809 | (1:64,000) |
| Antibody | Anti-Hoxc10 (Rabbit polyclonal) | PMID:31141687 | | (1:64,000) |
| Antibody | Anti-Hoxd10 (Guinea pig polyclonal) | Abcam | Cat# ab172865 | (1:16,000) |
| Antibody | Anti-Nos (Rabbit polyclonal) | Immunostar | Cat# 24431, RRID:AB_572255 | (1:10,000) |

*Continued on next page*

*Continued*

| Reagent type (species) or resource | Designation | Source or reference | Identifiers | Additional information |
|---|---|---|---|---|
| Antibody | Anti-Raldh2 (Guinea pig polyclonal) | PMID:18662545 | RRID: AB_2631299 | (1:32,000) |
| Antibody | Anti-Scip (Rabbit polyclonal) | PMID:28190640 | RRID:AB_2631304 | (1:8000) |
| Antibody | Anti-Lhx3 (Rabbit polyclonal) | Jessell lab | | (1:16,000) |
| Antibody | Anti-Mnx1 (Mouse monoclonal) | DSHB | Cat# MNR2, RRID: AB_2314625 | (1:100) |
| Antibody | Anti-GFP (Rabbit polyclonal) | Invitrogen | Cat# A-6455, RRID: AB_221570 | (1:5000) |
| Antibody | Anti-DIG AP | Sigma-Aldrich | Cat# 11093274910, RRID: AB_2734716 | (1:5000) |
| Recombinant DNA reagent | pGEM-Mnx1-Rybp (plasmid) | This paper | | See Materials and methods, and *Figure 1U* legend |
| Recombinant DNA reagent | pGEM-Mnx1-Cbx2 (plasmid) | This paper | | See Materials and methods, and *Figure 1U* legend |
| Recombinant DNA reagent | pCAGGS-mouseHoxa13-IRES2-nucEGFP (plasmid) | This paper | | See Materials and methods, and *Figure 7B* legend |
| Recombinant DNA reagent | pCAGGS-mouseHoxb13-IRES2-nucEGFP (plasmid) | This paper | | See Materials and methods, and *Figure 7B and D* legend |
| Commercial assay or kit | DIG RNA Labeling Kit | Sigma-Aldrich | Cat# 11175025910 | |
| Commercial assay or kit | One Taq One-Step RT-PCR | NEB | Cat# E5315S | |
| Commercial assay or kit | SMARTer Stranded RNA-Seq Kit | Takara | Cat# 634,837 | |
| Commercial assay or kit | Nextera XT DNA Library Preparation Kit | Illumina | Cat# FC-121–1,030 | |
| Commercial assay or kit | Papain Dissociation System | Worthington | PDS | |
| Commercial assay or kit | Arcturus Picopure RNA Isolation Kit | Applied Biosystems | KIT0204 | |
| Commercial assay or kit | NEBNext High-Fidelity 2 X PCR Master Mix | NEB | M0541S | |
| Commercial assay or kit | Qiagen MinElute | Qiagen | 28,204 | |
| Chemical compound, drug | Glycergel | Agilent | Cat# C0563 | |
| Chemical compound, drug | AMPure XP | Beckman Coulter Life Sciences | A63881 | |
| Chemical compound, drug | Turbo DNase | Invitrogen | AM2238 | |
| Chemical compound, drug | SYBR Green I Nucleic Acid Gel Stain | Invitrogen | S7563 | |
| Software, algorithm | Integrative Genomics Viewer | | RRID: SCR_011793 | https://software.broadinstitute.org/software/igv/ |
| Software, algorithm | Picard | | RRID: SCR_006525 | |
| Software, algorithm | BEDTools | | RRID:SCR_006646 | https://github.com/arq5x/bedtools2, *Quinlan, 2021* |
| Software, algorithm | GraphPad Prism | GraphPad Software | RRID: SCR_002798 | https://www.graphpad.com/ |
| Software, algorithm | MACS | | RRID: SCR_013291 | https://github.com/macs3-project/MACS, *Liu, 2021* |

## Mouse genetics

*Ezh1*[flox/flox] (*Hidalgo et al., 2012*), *Ezh2*[flox/flox] (*Su et al., 2003*), *Olig2*[Cre] (*Dessaud et al., 2007*), *Mnx1*[GFP] (*Arber et al., 1999*) mice have been previously described. *Rybp*[flox/flox]*::Yaf2*[-/-] (*Hisada et al., 2012*; *Rose et al., 2016*) mice were generated by microinjection of the mouse ESCs with these alleles into blastocysts followed by implantation into pseudopregnant female mice. Generation of Ezh[MNΔ] mice

was performed by crossing *Ezh1^flox/flox^*, *Ezh2^flox/flox^* and *Olig2^Cre^* mice. Ezh^MNΔ^ mice are viable at birth but do not survive beyond P20. Generation of Ring1^MNΔ^ mice was performed by crossing *Ring1^-/-^::Rnf2^flox/flox^* and *Olig2^Cre^*. Ring1^MNΔ^ mice perish at birth. Generation of Ring1^MNΔ^::*Mnx1-GFP* mice was performed by crossing *Ring1^-/-^::Rnf2 ^flox/flox^*, *Olig2^Cre^* mice to *Mnx1-GFP* mice. Animal work was approved by the Institutional Animal Care and use Committee of the NYU School of Medicine in accordance to NIH guidelines (Protocol IA16-00045).

## Slide immunohistochemistry

Embryos were fixed in 4% PFA for 1.5–2 hr at 4 °C, washed 5–6 times in cold PBS for 15–30 minutes each wash, and incubated overnight in 30% sucrose. Tissue was embedded in OCT, frozen in dry ice, and sectioned at 16 µm on a cryostat. For antibody staining of sections, slides of cryosections were placed in PBS for 5 min to remove OCT. Sections were then transferred to humidified trays and blocked for 20–30 min in 0.75 ml/slide of PBT (PBS, 0.1% Triton) containing 1% Bovine serum albumin (BSA). The blocking solution was replaced with primary staining solution containing antibodies diluted in PBT with 0.1% BSA. Primary antibody staining was performed overnight at 4 °C. Slides were then washed three times for 5 min each in PBT. Fluorophore-conjugated secondary antibodies were diluted 1:500-1:1000 in PBT and filtered through a 0.2 µm syringe filter. Secondary antibody solution was added to slides (0.75 ml/slide) and incubated at room temperature for 1 hr. Slides were washed three times in PBT, followed by a final wash in PBS. Coverslips were placed on slides using 110 µl of Vectashield (Vector Laboratories).

Antibodies against Hox proteins and MN subtypes were generated as described (*Dasen et al., 2008*; *Dasen et al., 2005*). Additional antibodies were used as follows: goat anti-Rnf2 (Abcam, 1:2000), rabbit anti-Rnf2 (Abcam, 1:5000), rabbit anti-Rybp (Abcam1:2000), rabbit anti-Yaf2 (Abcam, 1:2000), rabbit anti-Cbx2 (Bethyl, 1:5000), rabbit anti-H3K27me3 (Cell Signaling, 1:2000).

## In situ mRNA hybridization

Probe templates were generated by RT-PCR and incorporated a T7 promoter sequence in the antisense strand. Total RNA was first extracted from eviscerated E12.5 embryos using TRIzol (Invitrogen). Genes of interest were amplified with the One Taq One-Step RT-PCR kit (NEB) using 1 µg of RNA. After amplification by RT-PCR, a second PCR was performed to incorporate a T7 promoter sequence. Antisense riboprobes were generated using the Digoxigenin-dUTP (SP6/T7) labeling kit (Sigma-Aldrich). For in situ hybridization, sections were first dried for 10–15 min at room temperature, placed in 4% PFA, and fixed for 10 min at room temperature. Slides were then washed three times for 3 min each in PBS, and then placed in Proteinase K solution (1 µg/ml) for 5 min at room temperature. After an additional PFA fixation and washing step, slides were treated in triethanolamine for 10 min, to block positive charges in tissue. Slides were then washed three times in PBS and blocked for 2–3 hr in hybridization solution (50% formamide, 5 X SSC, 5 X Denhardt's solution, 0.2 mg/ml yeast RNA, 0.1 mg/ml salmon sperm DNA). Prehybridization solution was removed, and replaced with 100 µl of hybridization solution containing 100 ng of DIG-labeled antisense probe. Slides were then incubated overnight (12–16 hr) at 72 °C. After hybridization, slides were transferred to a container with 400 ml of 5 X SSC and incubated at 72 °C for 20 min. During this step, coverslips were removed using forceps. Slides were then washed in 400 ml of 0.2 X SSC for 1 hr at 72 °C. Slides were transferred to buffer B1 (0.1 M Tris pH 7.5, 150 mM NaCl) and incubated for 5 min at room temperature. Slides were then transferred to staining trays and blocked in 0.75 ml/slide of B1 containing 10% heat inactivated goat serum. The blocking solution was removed and replaced with antibody solution containing 1% heat inactivated goat serum and a 1:5000 dilution of anti-DIG-AP antibody (Roche). Slides were then incubated overnight at 4 °C in a humidified chamber. The following day, slides were washed three times, 5 min each, with 0.75 ml/slide of buffer B1. Slides were then transferred to buffer B3 (0.1 M Tris pH 9.5, 100 mM NaCl, 50 mM MgCl$_2$) and incubated for 5 min. Slides were then developed in 0.75 ml/slide of B3 solution containing 3.5 µl/ml BCIP and 3.5 µl/ml NBT for 12–48 hr. After color development, slides were washed in ddH$_2$0 and coverslipped in Glycergel (Agilent). A more detailed in situ hybridization protocol is available on our lab website (http://www.dasenlab.com).

## Wholemount immunohistochemistry

For wholemount immunohistochemistry embryos were fixed in PFA for 2 hr, then bleached for 24 hr at 4 °C in a 10% $H_2O_2$, 10% DMSO solution prepared in methanol. Embryos were washed three times for 10 min each in methanol, followed by five washes for 10 min in PBS. Primary antibodies were diluted in staining solution (5% BSA, 20% DMSO in PBS) and specimens were incubated in staining solution on a rotator overnight at room temperature. Samples were then washed three times for 5 min each in PBS, followed by four 1 hr washes in PBS. Specimens were then incubated in secondary antibodies diluted in staining solution overnight at room temperature. Samples were then washed three times for 5 min each in PBS, followed by four 1 hr washes in PBS, a single 10 min wash in 50% methanol, and three 20 min washes in 100% methanol. Samples were transferred to glass depression slides and tissue was cleared by incubating samples in BABB solution (1-part benzyl alcohol: 2-parts benzyl benzoate). Confocal images of embryos were obtained from Z-stacks using Zen software (Zeiss). Further details of wholemount staining protocols are available on our lab website: (http://www.dasenlab.com).

## In ovo chick electroporation

In ovo electroporation were performed on Hamburger Hamilton (HH) stage 13–14 chick embryos and analyzed at HH stage 24–25. Fertilized chicken eggs (Charles River) were incubated in a humidified incubator at 39 °C for 40–48 hr until they reached HH13-14. The top of the egg shell was removed and a 1 µg/µl DNA (150–500 ng/µL expression plasmid and pBKS carrier DNA) containing ~0.02% Fast green was injected into the central canal of the neural tube using a sharpened glass capillary tube. Electrodes (Platinum/Iridium (80%/20%), 250 µm diameter, UEPMGBVNXNND, FHC Inc) were placed on both sides of the neural tube (4 mm separation) and DNA was electroporated using an ECM 830 electroporator (ECM 830, BTX; 25 V, 4 pulses, 50ms duration, 1 s interval). Eggs were sealed with parafilm and incubated for 48 hr prior to fixation. Results shown in figures are representative of at least three electroporated embryos from two or more experiments in which electroporation efficiency in MNs was above 60%.

## RNA preparation and library preparation

RNA was extracted from FACS purified MNs dissected from E12.5 mouse embryo (10–20,000 cells/segment), using the Arcturus Picopure RNA isolation kit. For on-column DNase treatment, Turbo DNase was used (ambion, AM2238). Each samples used separated bar code for libraries. RNA quality and quantity were measured with an Agilent Picochip using a Bioanalyzer, all samples had quality scores between 9–10 RIN. For library preparation 10 ng of total RNA was used to generate cDNA, which was amplified with SMARTer Stranded RNA-Seq kit. 100 ng of cDNA were used as input to prepare the libraries (Takara, #634837), and amplified by 10 PCR cycles. Samples were run in four 50-nucleotide paired-end read rapid run flow cell lanes with the Illumina HiSeq 4000 sequencer.

## RNAseq data analyses

Sequencing reads were mapped to the assembled reference genome (mm10) using the STAR aligner (v2.5.0c) (*Dobin et al., 2013*). Alignments were guided by a Gene Transfer Format (GTF) file. The mean read insert sizes and their standard deviations were calculated using Picard tools (v.1.126) (http://broadinstitute.github.io/picard; *Broad Institute, 2019*). The read count tables were generated using HTSeq (v0.6.0) (*Anders et al., 2015*), normalized based on their library size factors using DEseq2 (*Love et al., 2014*), and differential expression analysis was performed. The Read Per Million (RPM) normalized BigWig files were generated using BEDTools (v2.17.0) (*Quinlan and Hall, 2010*) and bedGraphToBigWig tool (v4). To compare the level of similarity among the samples and their replicates, we used two methods: principal-component analysis and Euclidean distance-based sample clustering. All the downstream statistical analyses and generating plots were performed in R environment (v3.1.1) (https://www.r-project.org/).

## ATACseq

ATACseq was performed following previously described protocols (*Buenrostro et al., 2015*). DNA was extracted from purified dissected mouse embryonic MNs. Cells were aliquoted and washed twice in ice-cold 1× PBS. Cell pellets were resuspended in 10 mM Tris (pH 7.4), 10 mM NaCl, 3 mM MgCl2, 0.1% NP-40 (v/v), 0.1% tween20, 0.01% Digitonin and 1% BSA, centrifuged at 500 g for 5 min at

4 °C. Pellets were resuspended in 12.5 µl of 2× tagmentation DNA buffer, 1.25 µl Tn5 (Nextera DNA Sample Preparation Kit, FC-121–1030) and 11.25 µl of water, and incubated at 37 °C for 30 min. The sample was purified using the MinElute PCR Purification Kit (Qiagen, 28004). PCR enrichment of the library was performed with custom-designed primers and 2× NEB Master Mix. A qPCR reaction with 1× SYBR Green (Invitrogen), custom-designed primers and 2× NEB Master Mix (New England Labs, M0541) was performed to determine the optimal number of PCR cycles (one third of the maximum measured fluorescence) (*Buenrostro et al., 2013*). The libraries were purified using the AMPure XP beads (Beckman Coulter, A63880). High Sensitivity DNA ScreenTape (Agilent, 5067–5584) was used to verify the fragment length distribution of the library. Library quantification was performed using the KAPA Library Amplification kit on a Roche LightCycler 480. The libraries were sequenced on an Illumina NovaSeq (100 cycles, paired-end).

## ATACseq data analysis

All of the reads from the Sequencing experiment were mapped to the reference genome (mm10) using the Bowtie2 (v2.2.4)(*Langmead and Salzberg, 2012*) and duplicate reads were removed using Picard tools (v.1.126) (http://broadinstitute.github.io/picard/). Low-quality mapped reads (MQ < 20) were removed from the analysis. The read per million (RPM) normalized BigWig files were generated using BEDTools (v.2.17.0) (*Quinlan and Hall, 2010*) and the bedGraphToBigWig tool (v.4). Peak calling was performed using MACS (v1.4.2)(*Zhang et al., 2008*) and peak count tables were created using BEDTools. Differential peak analysis was performed using DESeq2 (*Love et al., 2014*). ChIPseeker (v1.8.0) (*Yu et al., 2015*) R package was used for peak annotations and motif discovery was performed using HOMER (v4.10)(*Heinz et al., 2010*). ngs.plot (v2.47) and ChIPseeker were used for TSS site visualizations and quality controls. To compare the level of similarity among the samples and their replicates, we used two methods: principal-component analysis and Euclidean distance-based sample clustering. The downstream statistical analyses and generating plots were performed in R environment (v3.1.1) (https://www.r-project.org/).

## Statistics

Samples sizes were determined based on previous experience and the number of animals and definitions of N are indicated in the main text and figure legends. In figures where a single representative image is shown, results are representative of at least two independent experiments, unless otherwise noted. No power analysis was employed, but sample sizes are comparable to those typically used in the field. Data collection and analysis were not blind. Graphs of quantitative data are plotted as means with standard error of mean (SEM) as error bars, using Prism 8 (Graphpad) software. Unless noted otherwise, significance was determined using unpaired t-test in Prism eight software, or using adjusted p-values. Exact p-values are indicated, where appropriate, in the main text, figures, and figure legends.

## Acknowledgements

We thank Kristen D'Elia, Sara Fenstermacher, Jessica Treisman, and Ed Ziff for discussion and comments on the manuscript, and Rachel Kim and Orly Wapinski for assistance. We thank Isabel Hidalgo and Susana Gonzalez for providing *Ezh1*flox/flox mice, Alexander Tarakhovsky for *Ezh2*flox/flox mice, Stefan Thor for *Eed* mutant embryos, Haruhiko Koseki for *Ring1*-/-::*Rnf2*flox/flox ES cells, and Robert Klose for *Ryby*flox/flox::*Yaf2*-/- ES cells. We also thank Genome Technology Center and Cytometry and Cell Sorting Laboratory at NYU Langone. This work was supported by NIH NINDS grants T32 GM007238, F31 NS087772 to AS, R01 NS 100897 to EOM, R35 NS116858, R01 NS062822 and R01 NS097550 to JD.

## Additional information

### Funding

| Funder | Grant reference number | Author |
|---|---|---|
| National Institutes of Health | R35 NS116858 | Jeremy S Dasen |
| National Institutes of Health | R01 NS062822 | Jeremy S Dasen |
| National Institutes of Health | R01 NS097550 | Jeremy S Dasen |
| National Institutes of Health | NS 100897 | Esteban Orlando Mazzoni |
| National Institutes of Health | T32 GM007238 | Ayana Sawai |
| National Institutes of Health | F31 NS087772 | Ayana Sawai |

The funders had no role in study design, data collection and interpretation, or the decision to submit the work for publication.

### Author contributions

Ayana Sawai, Conceptualization, Formal analysis, Investigation, Methodology, Writing - original draft, Writing – review and editing; Sarah Pfennig, Formal analysis, Investigation, Validation, Writing – review and editing; Milica Bulajić, Designed, performed, and analyzed mouse ESC experiments., Designed, performed, and analyzed mouse ESC experiments., Formal analysis, Investigation; Alexander Miller, Formal analysis, Investigation, Writing – review and editing; Alireza Khodadadi-Jamayran, Formal analysis, Methodology, Provided bioinformatic support with ATACseq and RNAseq data., Software; Esteban O Mazzoni, Designed, performed, and analyzed mouse ESC experiments., Designed, performed, and analyzed mouse ESC experiments., Funding acquisition, Methodology, Writing – review and editing; Jeremy S Dasen, Conceptualization, Funding acquisition, Supervision, Writing - original draft, Writing – review and editing

### Author ORCIDs

Ayana Sawai (iD) http://orcid.org/0000-0002-5446-4930
Esteban O Mazzoni (iD) http://orcid.org/0000-0001-8994-681X
Jeremy S Dasen (iD) http://orcid.org/0000-0002-9434-874X

### Ethics

Animals work was performed in strict accordance with the recommendations in the Guide for the Care and Use of Laboratory Animals of the National Institutes of Health. Animal work was approved by the Institutional Animal Care and use Committee of the NYU School of Medicine in accordance to NIH guidelines.

### Decision letter and Author response

Decision letter https://doi.org/10.7554/eLife.72769.sa1
Author response https://doi.org/10.7554/eLife.72769.sa2

## Additional files

### Supplementary files

• Supplementary file 1. DeSeq2 counts of RNA isolated from purified MNs of control and *Ring1* mutant embryos.

• Supplementary file 2. ATACseq counts from purified MNs of control and *Ring1* mutant embryos.

• Transparent reporting form

## Data availability

RNAseq and ATACseq data are available through GEO (GSE175503).

The following dataset was generated:

| Author(s) | Year | Dataset title | Dataset URL | Database and Identifier |
|---|---|---|---|---|
| Dasen JS | 2022 | PRC1 sustains the integrity of neural fate in the absence of PRC2 function | http://www.ncbi.nlm.nih.gov/geo/query/acc.cgi?acc=GSE175503 | NCBI Gene Expression Omnibus, GSE175503 |

The following previously published datasets were used:

| Author(s) | Year | Dataset title | Dataset URL | Database and Identifier |
|---|---|---|---|---|
| Bonev B | 2017 | Multi-scale 3D genome rewiring during mouse neural development | https://www.ncbi.nlm.nih.gov/geo/query/acc.cgi?acc=GSE96107 | NCBI Gene Expression Omnibus, GSE96107 |

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
