## [Editor Report]

This is an exciting and very well executed study, which will be of broad interest to the field of neuronal development. The demonstration of a similar logic in mouse as to what was reported earlier for *Drosophila* PRCs supports the idea of a deeply conserved mechanism of rostrocaudal patterning where PRC complexes control Hox gene expression.

---

## [Decision Letter]

**Decision letter after peer review:**

Thank you for submitting your article "PRC1 Sustains the Memory of Neuronal Fate Independent of PRC2 Function" for consideration by *eLife*. Your article has been reviewed by 3 peer reviewers, including Paschalis Kratsios as the Reviewing Editor and Reviewer #1, and the evaluation has been overseen by Marianne Bronner as the Senior Editor.

Essential revisions:

1. Additional experiments are necessary to better characterize the neurons in Ring1 mutant mice and address the following points: (a) it is unclear whether brachial motor neurons partially switch fate, adopt a mixed identity, and/or convert to a progenitor fate, and (b) the expected expansion of caudal motor neuron fates is not shown or quantified.

2. The authors are encouraged to include new experiments to strengthen the important point of PRC2 not being relevant and discuss the possible genetic perdurance of the PRC2 complex. Conclusive proof that the PRC2 (Ezh) knockout does not maintain persistent expression due to long half-life of mRNA, protein or H3K27me3 product is not provided. The published effect on later-born glia suggests that there might indeed be residual Ezh mRNA, protein or H3K27me3 product retained for some time after Cre expression. The authors may wish to test whether Hox13 genes are de-repressed in brachial domains of the PRC2 mutant. Moreover, the timing and potential weaknesses of the PRC2 genetic experiments should be discussed.

3. A deeper discussion should be included of what the authors think the H3K27me3 modification, which is apparently removed in their PRC2 depletions, might be doing in motor neurons. Lastly, the authors might wish to examine H3K27me3 levels at Hox loci prior to neurogenesis, if available mESC/progenitor datasets already exist.

*Reviewer #1:*

In vertebrates, much of our understanding of PRC functions comes from biochemical studies, mostly performed in vitro. As such, while we begin to understand the diverse mechanisms of PRC-dependent gene regulation, how PRC controls aspects of neuronal development remains largely unknown. In this manuscript, Sawai, Dasen and colleagues propose that PRC1 acts independently of de novo PRC2-dependent histone methylation to control motor neuron subtype specification. They perform a series of elegant genetic experiments in mice in which key PRC1 or PRC2 components are deleted from MN progenitors. The main conclusions of this current study are (1) PRC2 is dispensable for spinal motor neuron specification; (2) PRC1 is essential for the specification of segmentally-restricted motor neuron subtypes; (3) Canonical PRC1 regulates Hox gene expression and represses a variety of cell fate determinants; (4) Selective derepression of caudal Hox genes is associated with increased chromatin accessibility; (5) Hox13 paralogs repress rostral Hox genes by engaging accessible chromatin domains. This latter point is perhaps the most interesting conclusion of this paper as it provides mechanistic insights that, at least partially, explain the observed molecular changes in Ring1 MNDelta mice.

This is an exciting and very well executed study, which will be of broad interest to the field of neuronal development. Their demonstration of a similar logic in mouse as to what was reported earlier for *Drosophila* PRCs supports the idea of a deeply conserved mechanism of rostrocaudal patterning where PRCs control Hox gene expression. Strengths of this study include (1) the employment of a large collection of conditional mouse mutant lines; (2) the detailed transcriptomics analysis coupled with in vivo ATAC-Seq analysis, (3) the integration of in vitro data from mESC-derived MNs with in vivo experiments in mouse and chick embryos.

Overall, the conclusions in this paper are well supported and of high quality. One of the authors' main conclusions is that class-specific features of MNs are preserved in Ring1 MNDelta mice. However, the ectopic expression of various cell fate markers begs the question of whether some spinal MNs convert to a MN progenitor fate or adopt a mixed identity. The authors might wish to either further clarify this point, or test these possibilities with additional experiments. Lastly, the authors may want to consider testing Hox13 paralog expression in their PRC2 mutants to strengthen their conclusions.

The authors observe that class-specific features of MNs are preserved, such as neurotransmitter identity and Hb9 expression. However, strong axon innervation defects and ectopic expression of various cell fate markers are observed in Ring1 MNDelta mice, begging the question of whether some (not all) spinal MNs convert to a MN progenitor fate. Given the precedence in the literature of PRC affecting early patterning events and the observed increase in HMC progenitor cells (Figure 2H), experiments to address this question could strengthen the paper. For example, are markers for the MN progenitor domain (Olig2, Nkx6.1/2, Lhx3, Ngn1/2) expressed in the "presumptive MNs" of Ring1 MNDelta mice? Another intriguing possibility is that some spinal MNs in Ring1 MNDelta mice adopt a "mixed" identity since RNA-Seq analysis revealed ectopic expression of TFs normally involved in interneuron/regional neuronal specification (Figure 3B).

A previous study by the same lab (Golden and Dasen, G and D, 2012) manipulated the PRC1 component Bmi/Pcgf4 in spinal motor neurons. This study reached similar conclusions (e.g., Bmi1/Pcgf4 is dispensable for generating MNs as a class, Bmi1 controls rostral Hox gene boundaries), but Bmi1 gene manipulation resulted in a switch in MN fates. Given the ectopic expression of a variety of cell fate determinants (incl. numerous transcription factors) in spinal MNs of Ring1 mutant mice (lines 213-222), is there a possibility for a partial cell fate switch? The observed increase in HMC neurons and the presence of thicker axial MN projections support this possibility (Figure 2H), but this is not clearly articulated in the text. Lastly, it could help the reader if the Discussion includes a paragraph that compares the conclusions of the Golden and Dasen, G and D (2012) paper with the conclusions of the present study.

The authors observe robust derepression of Hox13 genes in all three segmental levels of the Ring1 MNDelta spinal cord. Their mechanistic experiments suggest that Hox13 is one of the main drivers of the observed phenotypes in Ring1 MNDelta mice by repressing expression of rostral Hox genes. It would be valuable to examine Hox13 expression levels in mice lacking PRC2 components to solidify one key conclusion of this study, that is, PRC2 is dispensable for MN specification.

Lines 58-61, 105-106: The manuscript in Introduction and Discussion is focused on the PRC2-dependent manner (via H3K27me3) by which PRC1 is thought to repress gene expression. However, other mechanisms are possible too. For example, H2A ubiquitination-dependent functions of PRC1 (that are PRC2 independent) have been previously reported to repress gene expression (summarized in this review PMID: 31398486). Discussion of such functions would improve the manuscript.

Usage of the word "memory" in Title, Abstract and Discussion is confusing. Moreover, the word memory does not adequately describe the key findings of this paper, i.e., retention of motor neuron fate coupled with ectopic expression of various fate determinants. If the authors decide to keep it, they should explain better what exactly "memory" means in their system. Lines 82-83: "These findings indicate that PRC1 can preserve the memory of early patterning events in the CNS". This sentence is not clear.

Lines 310-311, 319-321: The authors do a remarkable job to get at mechanism by incorporating RNA-Seq, ATAC-Seq and ChIP-seq data with chick in ovo electroporations. To strengthen their proposed model (line 431), the authors might wish to examine H3K27me3 levels at Hox loci prior to neurogenesis, if available mESC/progenitor datasets already exist.

*Reviewer #2 :*

The Polycomb group of transcriptional regulators is best known for their role in maintaining Hox gene expression during *Drosophila* development. Biochemical analysis from many labs revealed that they are broadly grouped into two types of complexes, called PRC1 and PRC2. The traditional view is that PRC2 acts first, by being recruited to target genes and then depositing a methyl mark on K27 of H3. PRC1 then reads this mark and mediates and maintains transcriptional repression.

Using genetic methods to remove the activities of core components of both PRC1 and PRC2, the authors show that, at least when the cre drivers (Olig2 and Sox1) they use are active, PRC1, but not PRC2, is essential to maintain segmentally-restricted Hox genes along the rostral-caudal axis of the mouse spinal cord. Consistent with altered Hox gene expression, motor neuron subtype specification is also disrupted in the PRC1 mutant (Ring1). The authors go on to assess motor neuron transcriptomes and chromatin accessibility in the Ring1 depletion background.

Altogether, the paper provides a very interesting, complete and thorough analysis of the effects of removing PRC1 core component Ring1. The figures and data are well documented, and the conclusions are solid.

Perhaps the lack of an effect of removing PRC2 components is not as surprising as the authors suggest and could be more clearly and precisely described. For example, given the potential for genetic perdurance of the complex, the exact stage at which PRC2 activity is removed in these experiments isn't clear and could be discussed more thoroughly. It was also unclear whether the analogous manipulations (in terms of the transition form mitotic to post-mitotic cells) have been rigorously carried out in other systems. The loss of H3K27me3 in the motor neuron domain is a good demonstration that these manipulations are working, but again the timing of this loss is not known or discussed.

In addition to better outlining the timing and potential weaknesses of their PRC2 genetic experiments, I would like to see a better description of why they switched from Olig2:cre to Sox1:cre when testing the role of Ezh and Eed, respectively. Would swapping drivers help make the negative result more convincing? Are there even earlier drivers that could be used?

I would also like to see a deeper discussion of what the authors think the H3K27me3 modification, which is apparently removed in their PRC2 depletions, might be doing in motor neurons. Have H3K27me3 ChIPs been carried out? Have RNAseq measurements been carried out in this background? What, if any, is the relationship between those sites of histone modification and genes that change expression or positions of chromatin accessibility that do or do not change? Even if more data are not added to this paper, it would be interesting to hear the authors' speculations.

I'm also moderately puzzled as to why de-repression of caudal Hox genes doesn't typically result in strong repression of more rostral Hox genes. Are they co-expressed in the same cells? Despite the repression observed in the chick electroporation experiments, is cross-repression between Hox genes not occur to a significant degree in this system?

Is it surprising that the changes in gene expression in the Ring1 mutant are largely the same for all three spinal cord regions (e.g. the heat maps in figure 3)?

*Reviewer (#3):*

This paper is a technical gem. They use Cre lines to knock out core components of PRC1 or PRC2 and assay the effect on HOX gene expression along the rostral/caudal axis of the mouse embryo. In addition to measuring HOX gene expression, they also do scRNA-seq from different regions of the CNS as well as ATAC-seq to assay for open chromatin. The experiments are clean, well controlled, and convincingly presented. On the other hand, I am concerned about the novelty of the study. This is because they primarily assay HOX gene expression, and we know from decades of work that the PRC1/2 complexes repress posterior HOX gene expression to prevent ectopic anterior expression. So that result is beautifully shown, but conceptually already well known. Yet they don't spend enough time characterizing the effect on neurons. They show that in the absence of PRC1 there is a loss of rostral neuron numbers, but they don't assay for ectopic caudal neurons, or the effect of this neuronal fate switch on the function of the MNs.

In addition, they don't prove that PRC2 is not relevant, as indicated in the title/abstract. They don't assay for loss of Ezh protein in the ezh knockout, and their proxy H3K27me3 could be functional even at low levels. (Note the level of Ezh or H3K27me3 was not quantified anywhere I could find.) And the published effect on later-born glia suggests that there might indeed be residual Ezh mRNA, protein or H3K27me3 product retained for some time after Cre expression. Of course, proving that a manipulation removes ALL gene product is a difficult task, but needs to be taken more seriously here given the conclusion is in the Title.

This paper is a technical gem. They use Cre lines to knock out core components of PRC1 or PRC2 and assay the effect on HOX gene expression along the rostral/caudal axis of the mouse embryo. In addition to measuring HOX gene expression, they also do scRNA-seq from different regions of the CNS as well as ATAC-seq to assay for open chromatin. The experiments are clean, well controlled, and convincingly presented.

On the other hand, I am not convinced it should be published in *eLife*. This is because they primarily assay HOX gene expression, and we know from decades of work that the PRC1/2 complexes repress posterior HOX gene expression to prevent ectopic anterior expression. So that result is beautifully shown, but conceptually already well known.

Second, they don't spend enough time characterizing the effect on neurons. They show that in the absence of PRC1 there is a loss of rostral neuron numbers, but they don't assay for ectopic caudal neurons, or the effect of this neuronal fate switch on the function of the MNs.

Third, they don't prove that PRC2 is not relevant, as indicated in the title/abstract. They don't assay for loss of Ezh protein in the ezh knockout, and their proxy H3K27me3 could be functional even at low levels. (Note the level of Ezh or H3K27me3 was not quantified anywhere I could find.) And the published effect on later-born glia suggests that there might indeed be residual Ezh mRNA, protein or H3K27me3 product retained for some time after Cre expression. Of course, proving that a manipulation removes ALL gene product is a difficult task, but needs to be taken more seriously here given the conclusion is in the Title.

---

## [Author Response]

Essential revisions:1. Additional experiments are necessary to better characterize the neurons in Ring1 mutant mice and address the following points: (a) it is unclear whether brachial motor neurons partially switch fate, adopt a mixed identity, and/or convert to a progenitor fate, and (b) the expected expansion of caudal motor neuron fates is not shown or quantified.

(a) In the revision we have clarified what are the ultimate fates of MNs in *Ring1* mutants. We have analyzed markers for MN progenitors (Olig2 and Nkx6.1) in *Ring1* mutants and find they are not ectopically expressed by postmitotic cells. Of note, Nkx6.1 is also expressed by postmitotic limb-level MN pools (De Marco Garcia *Neuron* 2008), and this expression is lost in *Ring1* mutants, consistent with a loss of Hoxdependent MN pool identities. These data are now shown in the revised Figure 1—figure supplement 2A.

We also find little evidence that MNs adopt a hybrid class identity in *Ring1* mutants. This conclusion is supported by (1) the lack of any reduction in the majority of general MN markers (Mnx1, Isl1, Lhx3, as well neurotransmitter synthesis pathway genes), (2) no marked induction of alternate neurotransmitter fates (i.e. vGlut or vGad induction) as shown in the revised Figure 3—figure supplement 1C. We realize that some of the TFs that are ectopically expressed by MNs in *Ring1* mutants include determinants of ventral spinal interneuron fates (e.g. Chx10 in V2a, Gata2 in V2b, and En1 in V1 interneurons), and therefore one could argue the transcriptional profiles of MNs have shifted away from a “true” MN. Therefore, we have modified the discussion to incorporate this feature as follows:

“Despite the derepression of multiple fate determinants in the absence of PRC1 function, markers of MN class identity (e.g. Mnx1, Isl1/2, and Lhx3) are largely unchanged, and MNs do not appear to acquire features of alternate neuronal class fates, beyond expression of fate-defining transcription factors.”

(b) We have also further addressed the question of what are the identities of the remaining MN subtypes in *Ring1* mutants. Although we observe ectopic expression of *Hox13* genes in rostral segments of *Ring1* mutants, we believe our results are most consistent with the remaining brachial and thoracic MNs having the default Hoxindependent “axial” MN fate (i.e. HMC and MMC subtype identities). This is due to MNs having indeterminate Hox codes, due to derepression of multiple caudal *Hox* genes. In the revision we have included new analyses showing that brachial MNs co-express Hoxc9 and Hoxc10 in Figure 1—figure supplement 2B. These neurons also likely co-express *Hox13* genes, although we do not have an antibody to address this. The increase in the numbers of HMC neurons is quantified in Figure 2H.

Because *Hoxc13* is normally expressed in sacral segments, which consists predominantly of axial MNs that presumably innervate the tail, as well as a small population of PGC-like visceral MNs, it is possible that they express markers of sacral subtypes (through a posterior dominance-like mechanism). However, the absence of ectopic expression of markers for sacral visceral spinal MNs (NOS or Foxp1) argues against MNs acquiring caudal identities. To clarify these issues we have added the following paragraph to the revised discussion.

“Our results indicate that brachial MNs ectopically express Hox9-Hox13 paralogs in Ring1 mutants, while expression of Hox4-Hox8 genes is diminished. As a consequence, MNs fail to acquire a limb-innervating LMC molecular fate, and express markers indicative of MMC and HMC subtypes. These observations are consistent with a model in which the Hox code of MNs is scrambled in Ring1 mutants, reverting to the fate of ancestral axial subtypes. However, we cannot formally rule out the possibility that some MNs acquire a Hox13^+^ fate, although we did not observe expression of markers of sacral PGC identity (nNos, Foxp1).”

2. The authors are encouraged to include new experiments to strengthen the important point of PRC2 not being relevant and discuss the possible genetic perdurance of the PRC2 complex. Conclusive proof that the PRC2 (Ezh) knockout does not maintain persistent expression due to long half-life of mRNA, protein or H3K27me3 product is not provided. The published effect on later-born glia suggests that there might indeed be residual Ezh mRNA, protein or H3K27me3 product retained for some time after Cre expression. The authors may wish to test whether Hox13 genes are de-repressed in brachial domains of the PRC2 mutant. Moreover, the timing and potential weaknesses of the PRC2 genetic experiments should be discussed.

The conditional alleles we used target exons encoding the SET domain, required for methytransferase activity, and is located on the C-terminal region of the protein. Unfortunately, all of the commercial Abs that work in IHC against Ezh2 and Ezh1 target the N-terminus, and Ezh2 protein expression is still detected in our conditional mutants. This is in part why we chose to use H3K27me3 staining, as it is a functional readout of PRC2 enzymatic activity, and shows clear depletion from both progenitors and postmitotic neurons by E11.5. One caveat is is that the deletion of either *Ezh* or *Eed* affects only de-novo methylation on newly synthesized histones, therefore the reduction of H3K27me3 is only manifested after several rounds of cell divisions without PRC2 activity. In the revision we include quantification of the reduction of H3K27me3 levels, showing markedly reduced levels in *Ezh* mutants (Figure 1—figure supplement 1A). We also performed an additional analyses to determine the time point in which H3K27me3 levels are reduced, showing detectable H3K27me3 at E10.5 (Figure 1—figure supplement 1B). We performed in situ hybridization against *Hox13* genes (*Hoxc13* and *Hoxb13*) and find no ectopic expression in *Ezh* mutants at E12.5, and show these new results in Figure 5—figure supplement 2.

We also attempted to perform in situ hybridization with antisense probes against deleted regions in the *Ezh1* and *Ezh2* genes, but were unsuccessful in obtaining clear signals. We believe this is due to the deleted region being relative small (~300), and because the sequence is conserved with other SET domain-containing genes.

The reported effects of PRC2 on glia (also using *Olig2::Cre*) we believe further support the conclusion that the *Ezh* mutant alleles are effective at depleting function in a replication coupled manner. The differences between MN and glial defects likely reflects that glia are born at later stages of development than MNs, and replicate more times over a longer period, and therefore H3K27me3 is more reduced. In the revision, we have included further discussion about the timing and potential pitfalls of the PRC2 genetic manipulations, and the influence of perdurance of H3K27me3. We revised to discussion as follows:

“Our studies show that Hox boundaries are maintained after genetic removal of PRC2 function, which leads to diminished H3K27me3 at the time of MN differentiation. Mutation of Ezh genes or Eed depletes H3K27me3 from progenitors and postmitotic neurons by E11.5, without appreciably affecting MN generation, Hox expression, or downstream Hox effectors. As the axial identity of progenitor cells appears to be shaped prior to neurogenesis (Metzis et al., 2018), our PRC2 manipulations likely remove H3K27me3 after Polycomb repression has been initiated. Consistent with this idea, our analyses of Ezh mutant as well as previous analyses of Eed^NE^ mice showed detectable levels of H3K27me3 in the MN domain until E10.5, indicating that some H3K27me3 remains during the phase in which anterior Hox boundaries are established (Yaghmaeian Salmani et al., 2018).

How does PRC1 restrict expression of Hox genes and other fate determinants in the absence of PRC2 function? One plausible scenario is that residual H3K27me3 is transmitted through cell division in PRC2 mutants and is sufficient to recruit canonical PRC1 and maintain target gene repression. As cells divide, newly synthesized histones are presumably devoid of H3K27me3 in PRC2 mutants, leading to replication-coupled dilution of H3K27me3. During neural differentiation, the rate of cell division decreases (Kicheva et al., 2014; Wilcock et al., 2007), potentially limiting H3K27me3 depletion, and enabling PRC1 to bind at target loci, even in the absence of de novo H3K27 methylation. Consistent with this idea, recent studies on the effects of PRC2 depletion in intestinal stem cells indicate that ~40% of residual H3K27me3 can maintain PRC repression (Jadhav et al., 2020), although the extent of derepression varies with gene target.”

3. A deeper discussion should be included of what the authors think the H3K27me3 modification, which is apparently removed in their PRC2 depletions, might be doing in motor neurons. Lastly, the authors might wish to examine H3K27me3 levels at Hox loci prior to neurogenesis, if available mESC/progenitor datasets already exist.

We agree this is an important point which we have provided further clarification and discussion in the revision. We argue that PRC2 is essential in regulating *Hox* expression, but works predominantly at the earliest stages of development, likely prior to neurogenesis. This idea is supported by studies in both ES cells and ES cell-derived MNs, showing removal of PRC2 leads to ectopic *Hox* expression. Analyses of PRC2 function during early neural differentiation has been challenging to study in vivo, as early loss of PRC2 typically has broad impact on cell viability, and the driver we used to target neuroectoderm (*Sox1:Cre*) is the earliest known driver for gene deletion in neural progenitors.

In the revision we further emphasize the important early role of PRC2 in both the introduction and the discussion. Of note, *Hox* clusters are generally covered by H3K27me3 in ES cells, while the early patterning signals RA and FGF remove these marks from rostral and caudal *Hox* genes, respectively, in MN precursors. We have clarified these findings in the introduction with the following revised paragraph:

“Polycomb repression is initiated by PRC2, which methylates histone H3 at lysine-27 (H3K27me3), permitting recruitment of PRC1 through subunits that recognize this mark, leading to chromatin compaction at genes targeted for repression (Margueron and Reinberg, 2011; Schuettengruber et al., 2017). In ES cells, Hox gene clusters are initially covered by H3K27me3 and loss of PRC2 function leads to reduced PRC1 binding and ectopic Hox expression (Boyer et al., 2006). During embryonic development, H3K27me3 marks are progressively removed from Hox clusters, allowing for the temporal and spatial activation of more caudal Hox genes during axis extension (Soshnikova and Duboule 2009). The progressive removal of PRC2-associated histone marks is also recapitulated in ES cell-derived MNs (ESC-MNs), where RA functions to deplete H3K27me3 from rostral Hox1-5 genes, while FGF removes H3K27me3 from more caudal Hox genes through Cdx proteins (Mazzoni et al., 2013). Although loss of PRC2 affects the viability of ESC-MNs, a hypomorphic mutation in the PRC2 component Suz12 causes ectopic expression of Hox genes. Thus, during early phases of embryonic development, PRC2 appears to have a critical role in establishing the early profiles of Hox expression along the rostrocaudal axis.”

Reviewer #1:[…]The authors observe that class-specific features of MNs are preserved, such as neurotransmitter identity and Hb9 expression. However, strong axon innervation defects and ectopic expression of various cell fate markers are observed in Ring1 MNDelta mice, begging the question of whether some (not all) spinal MNs convert to a MN progenitor fate. Given the precedence in the literature of PRC affecting early patterning events and the observed increase in HMC progenitor cells (Figure 2H), experiments to address this question could strengthen the paper. For example, are markers for the MN progenitor domain (Olig2, Nkx6.1/2, Lhx3, Ngn1/2) expressed in the "presumptive MNs" of Ring1 MNDelta mice?

In the revised paper, we have analyzed expression of Olig2 and Nkx6.1 finding no ectopic expression in the presumptive MNs of *Ring1* mutants. As noted in Essential revision 1, expression of Nkx6.1 by motor pools is lost. These data are now shown in Figure 1—figure supplement 2A. Lhx3 is normally expressed by MMC neurons, and this pattern is largely unchanged, with the exception of a small reduction at brachial levels, and the presence of a population of MNs that co-express Lhx3, Isl1, and Mnx1 (Hb9), as shown and quantified in Figure 2. See also Essential revision 1.

Another intriguing possibility is that some spinal MNs in Ring1 MNDelta mice adopt a "mixed" identity since RNA-Seq analysis revealed ectopic expression of TFs normally involved in interneuron/regional neuronal specification (Figure 3B).

See response to Essential revision 1.

A previous study by the same lab (Golden and Dasen, G and D, 2012) manipulated the PRC1 component Bmi/Pcgf4 in spinal motor neurons. This study reached similar conclusions (e.g., Bmi1/Pcgf4 is dispensable for generating MNs as a class, Bmi1 controls rostral Hox gene boundaries), but Bmi1 gene manipulation resulted in a switch in MN fates. Given the ectopic expression of a variety of cell fate determinants (incl. numerous transcription factors) in spinal MNs of Ring1 mutant mice (lines 213-222), is there a possibility for a partial cell fate switch?

See response to Essential revision 1.

The observed increase in HMC neurons and the presence of thicker axial MN projections support this possibility (Figure 2H), but this is not clearly articulated in the text.

We have clarified this result in the paper by adding paragraph in the discussion (see Essential revision 1). We have also quantified the increase in axial nerve thickness. These results are shown in Figure 2—figure supplement 1G.

Lastly, it could help the reader if the Discussion includes a paragraph that compares the conclusions of the Golden and Dasen, G and D (2012) paper with the conclusions of the present study.

In the Golden paper we showed that after acute depletion of the PRC1 component Bmi1 there is a switch in brachial LMC neurons to a thoracic PGC identity (due to derepression of Hoxc9). By contrast, in *Ring1* mutants there is a derepression of multiple *Hox9-Hox13* genes, leading to a loss of LMC neurons, but no switch to a PGC identity. The remaining motor neurons in *Ring1* mutants appear to have an indeterminate Hox code, and are molecularly similar to axial MMC or HMC neurons.

The differences between these phenotypes likely reflect the timing of the manipulations. In the Golden paper, *Bmi1* is acutely deleted from MNs in chick after axial identity has been established, whereas in *Ring1* mutants, the gene is removed at the time Olig2 is activated. We have added the following paragraph to clarify these differences (the most relevant being the differences in Bmi1 loss mediated through dsRNA electroporation near the time of MN differentiation versus genetic excision from Olig2+ progenitors in the Ring1 mutant).

“We previously found that acute depletion of the canonical PRC1 component Bmi1 in chick leads to ectopic Hoxc9 expression in brachial segments and a switch of LMC neurons to a thoracic PGC fate (Golden and Dasen 2012). By contrast in Ring1 mutants, multiple caudal Hox genes (Hox9-Hox13) are derepressed, and the remaining MNs have an axial subtype fate. While the exact mechanisms of these difference are unclear, they could reflect differences in the timing and spatial extent of the manipulations. In Ring1 mutants PRC1 function is depleted from MN progenitors throughout the rostrocaudal axis, whereas Bmi1 knockdown was performed in brachial segments just prior to differentiation.”

The authors observe robust derepression of Hox13 genes in all three segmental levels of the Ring1 MNDelta spinal cord. Their mechanistic experiments suggest that Hox13 is one of the main drivers of the observed phenotypes in Ring1 MNDelta mice by repressing expression of rostral Hox genes. It would be valuable to examine Hox13 expression levels in mice lacking PRC2 components to solidify one key conclusion of this study, that is, PRC2 is dispensable for MN specification.

We thank the reviewer for this excellent suggesting and have now performed an analysis of *Hox13* (*Hoxc13* and *Hoxb13*) gene expression in *Ezh dKO* mice. Consistent with our analyses of Hox protein expression, we observe no increase in *Hox13* expression in *Ezh* mutants. These results are shown in Figure 5—figure supplement 2.

Lines 58-61, 105-106: The manuscript in Introduction and Discussion is focused on the PRC2-dependent manner (via H3K27me3) by which PRC1 is thought to repress gene expression. However, other mechanisms are possible too. For example, H2A ubiquitination-dependent functions of PRC1 (that are PRC2 independent) have been previously reported to repress gene expression (summarized in this review PMID: 31398486). Discussion of such functions would improve the manuscript.

We thank the reviewer for this suggestion. The role of PRC1-mediated H2A ubiquitination in the repression *Hox* genes is somewhat controversial, as studies in both mouse and fly have shown this enzymatic function is not required for repression/compaction of *Hox* genes in early development (PMID: 26178786, 26385961). Nevertheless, we have revised discussion to raise the possibility that this function may contribute to PRC2independent functions of PRC1 in MNs as follows:

“Stabilization of PRC1 at repressed loci could also maintain target repression independently of de novo H3K27 methylation. Repression by canonical PRC1 has been shown to depend on the formation of phase-separated subnuclear structures (Polycomb bodies) assembled through polymerization of Polyhomeotic-like (Phc) and/or Cbx proteins (Isono et al., 2013; Plys et al., 2019; Tsuboi et al., 2018). Mutation of Phc2 in mice leads to ectopic expression of Hox genes, with Hox13 genes among the most robustly derepressed targets (Isono et al., 2013). One possibility is that PcG-mediated repression may not exclusively depend on anchoring of PRC1 through H3K27me3, but is also maintained through Phc-mediated chromatin compaction at specific loci. Alternatively, PRC2-independent repression by PRC1 could be facilitated through its H2A-ubiquitination activity, as has been suggested in other systems (Tsuboi et al., 2019).”

Usage of the word "memory" in Title, Abstract and Discussion is confusing. Moreover, the word memory does not adequately describe the key findings of this paper, i.e., retention of motor neuron fate coupled with ectopic expression of various fate determinants. If the authors decide to keep it, they should explain better what exactly "memory" means in their system. Lines 82-83: "These findings indicate that PRC1 can preserve the memory of early patterning events in the CNS". This sentence is not clear.

We agree with the reviewer that this term is somewhat vague and does not completely encapsulate the changes we observe. We have therefore removed it from most places in the text, specifically in reference to our own results, and replaced it with more precise language. We have left “memory” in a few places (e.g the introduction) where we generally describe PRC function, as this is an accepted term to describe the inheritance of transcriptional identities through cell proliferation.

Lines 310-311, 319-321: The authors do a remarkable job to get at mechanism by incorporating RNA-Seq, ATAC-Seq and ChIP-seq data with chick in ovo electroporations. To strengthen their proposed model (line 431), the authors might wish to examine H3K27me3 levels at Hox loci prior to neurogenesis, if available mESC/progenitor datasets already exist.

This is a very good suggestion. In the introduction we further describe previous characterizations of the pattern of H3K27me3 in stem cells, in the developing embryo, and in ESC-derived MNs. These studies are in agreement with a model in which H3K27me3 marks are progressively removed from *Hox* loci during axis extension. Specific changes to the text are detailed in Essential revision 3.

Reviewer #2:The Polycomb group of transcriptional regulators is best known for their role in maintaining Hox gene expression during *Drosophila* development. Biochemical analysis from many labs revealed that they are broadly grouped into two types of complexes, called PRC1 and PRC2. The traditional view is that PRC2 acts first, by being recruited to target genes and then depositing a methyl mark on K27 of H3. PRC1 then reads this mark and mediates and maintains transcriptional repression.Using genetic methods to remove the activities of core components of both PRC1 and PRC2, the authors show that, at least when the cre drivers (Olig2 and Sox1) they use are active, PRC1, but not PRC2, is essential to maintain segmentally-restricted Hox genes along the rostral-caudal axis of the mouse spinal cord. Consistent with altered Hox gene expression, motor neuron subtype specification is also disrupted in the PRC1 mutant (Ring1). The authors go on to assess motor neuron transcriptomes and chromatin accessibility in the Ring1 depletion background.Altogether, the paper provides a very interesting, complete and thorough analysis of the effects of removing PRC1 core component Ring1. The figures and data are well documented, and the conclusions are solid.Perhaps the lack of an effect of removing PRC2 components is not as surprising as the authors suggest and could be more clearly and precisely described. For example, given the potential for genetic perdurance of the complex, the exact stage at which PRC2 activity is removed in these experiments isn't clear and could be discussed more thoroughly.

We thank the reviewer for this suggestion and have revised both the results and discussion accordingly. We have removed statements describing this finding as unexpected or surprising in the revision. Previous characterization of *Sox10:Cre;Eed* mice indicated that H3K27me3 is undetectable in the neural tube by E11.5, while it is still detectable at ~E10. We have also performed an additional analysis of *Ezh* mutant embryos, showing detectable H3K27me3 at E10.5. We have revised introduction and discussion to clarify 1) the evidence that PRC2 does play important roles at early stages (work mostly done in ES cells and ES cell-derived MNs and (2) that the absence of a *Hox* phenotype could be due to perdurance of this histone mark during MN differentiation (discussion). See Essential Revisions Essential revision 2 and Essential revision 3 for specific changes in the text.

It was also unclear whether the analogous manipulations (in terms of the transition form mitotic to post-mitotic cells) have been rigorously carried out in other systems. The loss of H3K27me3 in the motor neuron domain is a good demonstration that these manipulations are working, but again the timing of this loss is not known or discussed.

We have clarified these issues in the revision, by (1) citing recent work looking at the effects of replication coupled dilution of H3K27me3 in intestinal stem cells (Jadhav et al., Mol Cell 2020), and (2) performing additional analyses of *Ezh* mutants at earlier stages of development. Specific changes are detailed in Essential revision 2 and Essential revision 3.

In addition to better outlining the timing and potential weaknesses of their PRC2 genetic experiments, I would like to see a better description of why they switched from Olig2:cre to Sox1:cre when testing the role of Ezh and Eed, respectively. Would swapping drivers help make the negative result more convincing? Are there even earlier drivers that could be used?

We have clarified why we switched to *Sox1::Cre* in the revision. *Sox1* is the earliest expressed gene that we are aware of that specifically targets neuroectoderm, which was the original motivation for switching to this driver. We have added the following to the Results section to clarify:

“Removal of Ezh genes via Olig2^Cre^ depletes PRC2 function in MN progenitors, raising the possibility that PRC2 regulates Hox expression at earlier stages. We therefore analyzed mice in which the core PRC2 component Eed was deleted using Sox1^Cre^ (Eed^NEΔ^ mice), which targets Cre to neuroectoderm (Takashima et al., 2007; Yaghmaeian Salmani et al., 2018). In Eed^NEΔ^ mice, H3K27me3 was depleted from spinal progenitors and postmitotic neurons by E11.5, with some H3K27me3 present in the floor plate (Figure 1—figure supplement 1G). In Eed^NEΔ^ mice, the number of Mnx1^+^ cells, and pattern of Hoxc6, Hoxc9, and Hoxc10 in MNs were unaffected, similar to Ezh^MNΔ^ mice (Figure 1—figure supplement 1F,H-J). These observations indicate that depletion of PRC2 function does not affect MN class specification or rostrocaudal positional identities at the time of differentiation.”

I would also like to see a deeper discussion of what the authors think the H3K27me3 modification, which is apparently removed in their PRC2 depletions, might be doing in motor neurons. Have H3K27me3 ChIPs been carried out? Have RNAseq measurements been carried out in this background? What, if any, is the relationship between those sites of histone modification and genes that change expression or positions of chromatin accessibility that do or do not change? Even if more data are not added to this paper, it would be interesting to hear the authors’ speculations.

Previous studies using ChIP-seq to measure H3K27me3 at *Hox* loci have shown H3K27me3 marks are removed progressively from *Hox* clusters along the rostrocaudal axis during axis extension in mouse embryos, and loss of these marks is associated with *Hox* gene activation (Soshnikova and Duboule 2009). A similar progression in H3K27me3 removal is observed in ES-derived MNs treated with the patterning cues RA and FGF (Mazzoni et al., 2013). PRC2 mutants typically perish at preimplantation stages, which has limited studies of changes in gene expression during neural development in vivo. Nevertheless, RNAseq has been performed on brain regions of *Eed^NEΔ^* mutants (Yaghmaeian Salmani et al., 2018), showing derepression of *Hox* genes in the brain. We suspect that the differences between the brain and spinal cord may related to differences in the proliferation, as the brain generates a larger number of neurons, and where any residual H3K27me3 may be further diluted. We do believe PRC2/H3K27me3 is important at early stages of development, and the lack of *Hox* changes in PRC2 mutants could be due to perdurance of the histone mark. We have modified the discussion to incorporate these ideas. See Essential revision 2 and Essential revision 3 for specific changes in the revision.

I’m also moderately puzzled as to why de-repression of caudal Hox genes doesn’t typically result in strong repression of more rostral Hox genes.

There is some disconnection between what we observe with Hox Ab staining/in situ and the level of repression as measured by RNAseq. With Ab staining, expression of the brachial Hox proteins is diminished to undetectable levels in *Ring1* mutants. The in situ we performed for *Hoxc6* shows similar pronounced reduction in *Hox* expression, while the reduction measured by RNAseq DESeq2 counts is ~50%. The less pronounced reduction in the RNAseq results could be due contamination in the sorted neurons, which may include some interneurons. This possibility was a motivating factor in confirming both derepressed and repressed genes by in situ hybridization.

Are they co-expressed in the same cells? Despite the repression observed in the chick electroporation experiments, is cross-repression between Hox genes not occur to a significant degree in this system?

Yes, we do observe co-expression of Hoxc9 and Hoxc10 protein in brachial segments, although the absolute levels of both are lower than what we observe in their normal domains (thoracic and lumbar, respectively). Presumably, this reduction in Hoxc9 and Hoxc10 expression is due to repression by Hox13 proteins, via cross repression. Nevertheless, these results are consistent with the idea that in *Ring1* mutants the Hox code is scrambled. We have added this data to Figure 1—figure supplement 2B.

Is it surprising that the changes in gene expression in the Ring1 mutant are largely the same for all three spinal cord regions (e.g. the heat maps in figure 3)?

Yes, the changes in gene expression are similar in each region for derepressed genes. We think that given that majority of the derepressed genes encode transcription factors, most of which are not normally expressed by MNs, this result is not surprising. The DESeq2 counts shown for select genes in Figure 4 also show similar degrees of derepression in each segment. This result contrasts with some of the *Hox* genes which normally have restricted expressed in MNs, where we do observe some segment-specific expression changes (for example Hoxc10 is derepressed in brachial and thoracic segments, but not lumbar).

Reviewer #3:This paper is a technical gem. They use Cre lines to knock out core components of PRC1 or PRC2 and assay the effect on HOX gene expression along the rostral/caudal axis of the mouse embryo. In addition to measuring HOX gene expression, they also do scRNA-seq from different regions of the CNS as well as ATAC-seq to assay for open chromatin. The experiments are clean, well controlled, and convincingly presented.On the other hand, I am not convinced it should be published in eLife. This is because they primarily assay HOX gene expression, and we know from decades of work that the PRC1/2 complexes repress posterior HOX gene expression to prevent ectopic anterior expression. So that result is beautifully shown, but conceptually already well known.

We agree that the important roles of PRCs in regulating *Hox* expression are known from both classic studies in flies and worms, as well as many studies in mammalian systems (largely in vitro). We do believe though that the differences between PRC1 and PRC2 function have not been directly compared in an in vivo model, and their specific functions in vertebrate neuronal diversification and chromatin organization are poorly defined. We also think it is surprising there does not appear to be a large-scale derepression of gene expression in PRC1 mutants, and that class-specific features MNs are maintained. Also, as there has been much attention to variant PRC1 in recent years, we believe our analyses of Rybp mutants will be broadly interesting to those that work on PRCs in other developmental contexts.

Second, they don’t spend enough time characterizing the effect on neurons. They show that in the absence of PRC1 there is a loss of rostral neuron numbers, but they don’t assay for ectopic caudal neurons, or the effect of this neuronal fate switch on the function of the MNs.

In the revision we have clarified what are the ultimate fates of MNs in the absence of PRC1/Ring1 function. Our results indicate that MNs do not transform to a caudal fate, but are retained in the more ancestral “Hox-less” axial columnar identity, likely due to MNs having an ambiguous Hox code. See specific changes to the revised text and figures to clarify this point are detailed in Essential revision 1.

Third, they don’t prove that PRC2 is not relevant, as indicated in the title/abstract. They don’t assay for loss of Ezh protein in the ezh knockout, and their proxy H3K27me3 could be functional even at low levels. (Note the level of Ezh or H3K27me3 was not quantified anywhere I could find.) And the published effect on later-born glia suggests that there might indeed be residual Ezh mRNA, protein or H3K27me3 product retained for some time after Cre expression. Of course, proving that a manipulation removes ALL gene product is a difficult task, but needs to be taken more seriously here given the conclusion is in the Title.

We agree that even if PRC2 gene function is effectively depleted there is still the possibility of remaining H3K27me3, by virtue of inheritance from cells that had this modification prior to deletion. We have revised the paper and our interpretation of these results as detailed in Essential revision 1.

We appreciate the concerns of the reviewer about the title of the paper. It was not our intention to convey that PRC2 is not important, but rather that the enzymatic activity is not needed at the time of differentiation. We have revised the title to be more specific that we are referring to the function of the PRC2 encoding genes, not specifically about H3K27me3.

We have also clarified in the discussion that we are not arguing that PRC2 is never important for Hox gene regulation, but that it is likely operating to set the profile of PRC1 recruitment at Hox loci prior to neurogenesis, as suggested by Metzis et al. 2018. We have clarified this as detailed in Essential revision 3.